# Effect of helical kink in antimicrobial peptides on membrane pore formation

**Alzbeta Tuerkova[1,2†‡], Ivo Kabelka[1,2†], Tereza Králová[1], Lukáš Sukeník[1,3], Šárka Pokorná[4], Martin Hof[4], Robert Vácha[1,2,3]\***

[1]CEITEC – Central European Institute of Technology, Masaryk University, Kamenice, Czech Republic; [2]National Centre for Biomolecular Research, Faculty of Science, Masaryk University, Kamenice, Czech Republic; [3]Department of Condensed Matter Physics, Faculty of Science, Masaryk University, Kotlářská, Czech Republic; [4]J. Heyrovsky Institute of Physical Chemistry, Czech Academy of Sciences, Prague, Czech Republic

**Abstract** Every cell is protected by a semipermeable membrane. Peptides with the right properties, for example Antimicrobial peptides (AMPs), can disrupt this protective barrier by formation of leaky pores. Unfortunately, matching peptide properties with their ability to selectively form pores in bacterial membranes remains elusive. In particular, the proline/glycine kink in helical peptides was reported to both increase and decrease antimicrobial activity. We used computer simulations and fluorescence experiments to show that a kink in helices affects the formation of membrane pores by stabilizing toroidal pores but disrupting barrel-stave pores. The position of the proline/glycine kink in the sequence further controls the specific structure of toroidal pore. Moreover, we demonstrate that two helical peptides can form a kink-like connection with similar behavior as one long helical peptide with a kink. The provided molecular-level insight can be utilized for design and modification of pore-forming antibacterial peptides or toxins.

**\*For correspondence:**
robert.vacha@mail.muni.cz

[†]These authors contributed equally to this work

**Present address:** [‡]Department of Pharmaceutical Chemistry, University of Vienna, Vienna, Austria

**Competing interests:** The authors declare that no competing interests exist.

## Introduction

One of the greatest threats to global health is the emergence and spreading of bacterial strains resistant to antibiotics (*Fair and Tor, 2014*; *WHO, 2017*). Despite the many safety measures, resistant bacteria can be commonly found in clinical settings, causing the so-called nosocomial infections. Such infections are associated with higher hospital costs due to extended treatment, life-threatening conditions, and increased mortality. Each year, hundreds of thousands of people die after contracting an infection caused by drug-resistant bacteria (*WHO, 2017*). Moreover, infections by resistant bacteria are projected to become one of the leading causes of premature death in the upcoming decades. Unfortunately, the current rate of introducing novel antimicrobial drugs is very low and new classes of antibiotics have not been approved for decades (*Nelson et al., 2019*). Therefore, there is a need for the development of new types of pharmaceuticals.

One of the promising candidates for the treatment of antibiotic-resistant bacterial infections are antimicrobial peptides (AMPs). AMPs are naturally occurring peptides that comprise of both (1) host-defense peptides, i.e., components of innate immune system of many organisms (*Fosgerau and Hoffmann, 2015*; *Peters et al., 2010*) and (2) peptide toxins with antimicrobial activity (*Zasloff, 2002*). To this day, thousands of AMPs (*Wang et al., 2015*) were discovered and many of them exhibit their activity (against bacteria, viruses, fungi, or cancer cells) even at micromolar concentrations (*Gaspar et al., 2013*; *Falco et al., 2009*; *Hoskin and Ramamoorthy, 2008*). AMPs have not been adopted as widely as conventional antibiotics due to their common toxicity, increased tendency for degradation by proteases, high cost, and decreased activity in vivo (*Mahlapuu et al., 2016*). Additionally, the rational design is not feasible, because we lack the understanding of the

sequence-to-function relationship (*Zhou et al., 2016*). Even a single residue substitution can have profound effect on the peptide activity and its mechanism of action (*Zhou et al., 2016*).

A wide variety of mechanisms of action were proposed for AMPs (*Wang et al., 2015*). AMPs can be directly responsible for killing the pathogens or they can act as immunomodulators (*van der Does et al., 2019Zhou et al., 2016*). The direct mechanisms include (but are not limited to): (1) agglutination of the pathogens (*Kumar et al., 2016*), (2) translocation across the membrane and possible interaction with intracellular targets (*Park et al., 2000*; *Dougherty et al., 2019*), (3) total solubilization of the membranes (*Kumar et al., 2018*), and (4) pore formation (*Kumar et al., 2018*; *Guha et al., 2019*). The latter mechanism is the main focus of this work. Description of other mechanisms goes beyond the scope of this article and has been reviewed elsewhere (*Kumar et al., 2018*; *Guha et al., 2019*; *Dougherty et al., 2019*).

Common characteristics of pore-forming AMPs are: (1) amphiphilicity (spatial clustering of hydrophobic and hydrophilic residues) (*Wimley, 2010*), (2) positive net charge, and (3) length ranging from 10 to 50 amino acids (*Wang et al., 2015*). AMPs are typically unstructured in solution and can adopt a secondary structure upon interaction with a membrane. Further on, we consider the most common α-helical peptides. The effect of individual peptide residues on lipid specificity remains unclear, however, positive net charge of AMPs was shown to be responsible for increased selectivity towards bacterial membranes (*Chen et al., 2005*; *Dathe and Wieprecht, 1999*). The amount and the distribution of hydrophobic residues, which influence peptide self-association and binding into the membrane, can be conveniently represented by a helical wheel projection (*Schiffer and Edmundson, 1967*). For secondary amphiphilic peptides, hydrophobic content can be described as an angle of the sector in the helical wheel. The hydrophobic length (cluster of residues along the peptide long axis) of pore-forming peptides enables them to span the lipid bilayer and determines the peptide orientation within the pore based on hydrophobic mismatch (*Vácha and Frenkel, 2013*).

Pore-forming peptides can be further characterized by the structure of the pore they form. There are two well-established pore models: (1) barrel-stave and (2) toroidal. In both pore structures, hydrophobic residues of AMPs are in contact with the membrane core, while hydrophilic residues form a polar channel. Barrel-stave pore is a compact, bundle-like assembly of peptides with only a little effect on the neighboring lipids (*Pollard et al., 1995*). In contrast, peptides in toroidal pores are loosely arranged and the lipid headgroups are present in the polar channel (*Sengupta et al., 2008*). Due to the small size and transient nature of the pores, it is challenging to experimentally determine the structure of membrane pores. However, computational methods are able to capture such transient structures and provide molecular-level understanding of the underlying mechanism of action. Thus, considerable number of studies on pore formation employed computational methods (*Leontiadou et al., 2006*; *Sengupta et al., 2008*; *Illya and Deserno, 2008*; *Mihajlovic and Lazaridis, 2012*; *Santo et al., 2013*; *Vácha and Frenkel, 2013*; *Bennett et al., 2014*; *Kabelka and Vácha, 2015*; *Kirsch and Böckmann, 2016*; *Chen et al., 2019*; *Miyazaki et al., 2019*).

The topology of a pore structure is determined by peptide properties. Particularly interesting is the presence of proline or glycine residue in a peptide sequence, which can induce formation of a sharp bend (so-called kink) in the regular α-helical structure. The kink (*Vanhoof et al., 1995*; *Lee et al., 2013*) and helix-kink-helix motif (*Kozic et al., 2018*) have been shown to be biologically relevant for the activity of AMPs. However, methodologically diverse studies have produced contradictory results, reporting the helical kink to both enhance (*Xiao et al., 2009*; *Suh et al., 1999*; *Lim et al., 2005*; *Shin et al., 2001*; *Suh et al., 1996*; *Kobayashi et al., 2000*; *Tieleman et al., 2001*; *Rodríguez et al., 2014*; *Corzo et al., 2001*; *Vermeer et al., 2012*; *Amos et al., 2016*) and reduce (*Park et al., 2002*; *Gehman et al., 2008*; *Liu et al., 2011*) antimicrobial effects.

The main objective of this study is to determine and provide molecular understanding of the effect of proline/glycine-induced kink on the peptide pore-forming ability. Our findings were obtained by multi-scale computer simulations and validated by *in vitro* fluorescence experiments. Firstly, we performed Monte Carlo (MC) simulations with a mesoscale lipid model (*Cooke and Deserno, 2005*) together with a phenomenological coarse-grained model for amphiphilic helical peptides (*Vácha and Frenkel, 2011*). Both of these models were previously shown to capture important peptide properties for pore formation (*Vácha and Frenkel, 2013*; *Kabelka and Vácha, 2015*). The free energy calculations (performed under various conditions with the Wang-Landau method [*Wang and Landau, 2001*]) provided general explanation of the effect of a peptide kink. Secondly,

the conclusions drawn from the MC simulations were confirmed by molecular dynamics (MD) simulations by using a more detailed MARTINI model (*Monticelli et al., 2008*). Sequences of Magainin 2 (*Matsuzaki et al., 1995*), LL-37 (*Turner et al., 1998*), Buforin II (*Park et al., 1998*), δ-lysin (*Janzon et al., 1989*), Candidalysin (*Moyes et al., 2016*) were altered (single amino acid substitutions) to study the impact of the kink on the pore structure and stability. Both of these distinct models have consistently shown that the presence of a kink disrupts barrel-stave, but stabilizes toroidal pores. Moreover, the position of the kink (with respect to the hydrophobic surface on AMP) was identified as a molecular determinant of peptide arrangement in toroidal pores. Finally, the pore-forming activity of selected peptide variants was verified using fluorescence leakage assay on large unilamellar vesicles (LUVs). Collectively, this information can be used to fine-tune peptide sequences to modulate the peptide activity.

## Results

### Monte Carlo simulations with phenomenological model
#### Free energy of pore opening at various conditions

We have evaluated the effect of kink-induced flexibility on the stability of barrel-stave and toroidal pore structures. Using a phenomenological (highly coarse-grained) model of amphiphilic peptides, we have calculated the free energy associated with pore formation by peptides with various properties. We modified the peptide length, presence of the kink, hydrophobicity (capping) of the peptide termini, and the hydrophobic content defined as a sector in helical wheel projection (see the Methods section for more information).

The presence of kink was found to be unfavorable for the formation of the barrel-stave pore, because the (kink-induced) increased flexibility hinders the tight packing of peptides inside barrel-stave pores. This preference is demonstrated in *Figure 1A*, which captures the free energy of pore formation for both rigid and flexible (with kink) peptides with the length 5 nm and the hydrophobic content/sector of the 270°. Due to the high hydrophobic content, the flexible peptide

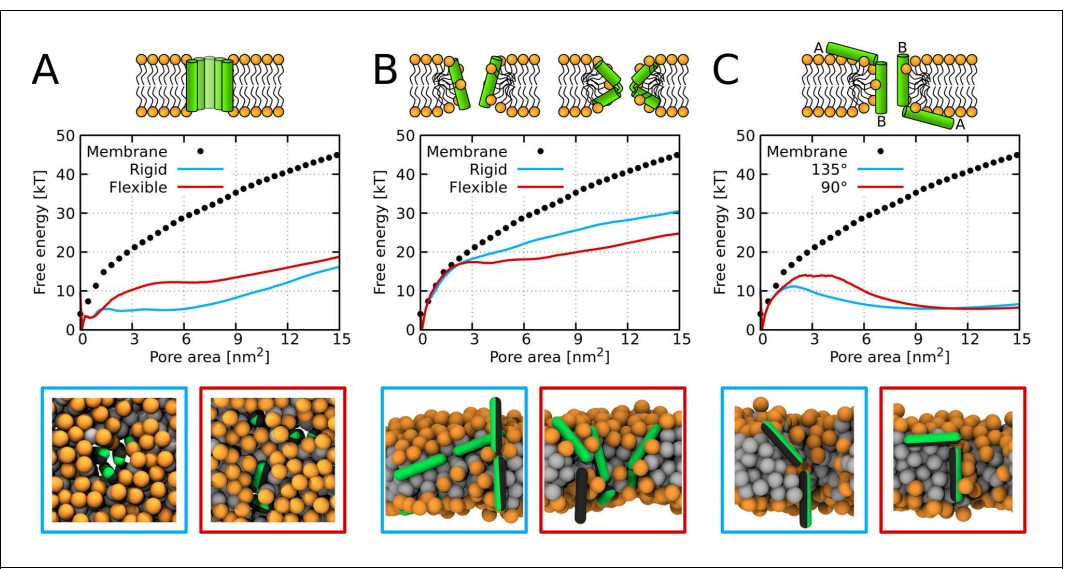

**Figure 1.** Free energy profiles of pore formation. (**A**) Barrel-stave pore, (**B**) toroidal pore, and (**C**) synergistic pore. Schematic images of studied pore structures (top) and representative simulation snapshots of the phenomenological model (bottom) are shown. Labels flexible and rigid correspond to peptides with and without a kink, respectively. The dotted line corresponds to the free energy profile of our reference system, a pure membrane without any peptides. Color coding: lipid headgroup = orange, lipid tail = light gray, hydrophilic = green, hydrophobic = dark gray.
The online version of this article includes the following source data for figure 1:

**Source data 1.** Free energy profiles and system configurations for snapshots.

variants can still insert into the membrane. However, the peptides are less prone to aggregation in membrane and consequently less likely to form pores. In contrast, peptides forming toroidal pores benefit from the kink-induced flexibility as can be seen in *Figure 1B*, where free energy profiles for half-hydrophobic peptides of length 7 nm are displayed. Additional examples of barrel-stave and toroidal pore forming peptides are shown in *Appendix 1—table 1*.

A pore can be also created by association between surface-aligned and transmembrane peptides. It has been suggested that a mixture of suitable AMPs could possess synergistic interactions responsible for more efficient pore formation (*Salnikov and Bechinger, 2011*). In our simulations, we have observed these synergistic pores created by half-hydrophobic peptides with the helix-kink-helix motif. Our peptide was comprised of two 3 nm long helices connected by a rigid kink, see *Figure 1C*. We have considered two cases where the angle between the peptides was 135° and 90°. In agreement with the experimental evidence (*Strandberg et al., 2016*), the calculated free energy profiles show the preference for the angle of 135° compared to 90°.

Furthermore, we have calculated the free energy of pore formation for half-hydrophobic (a hydrophobic sector of 180°) peptides. The following parameters were considered: (1) flexibility of peptide kink, (2) hydrophobicity of peptide termini, and (3) peptide concentration. Due to the low binding of some peptides on the membrane, we also considered peptides with an increased interaction strength, that is higher hydrophobic moment (see the Materials and methods section for more information). For peptides that were bound on the membrane at sufficiently high concentration, flexible kink was observed to be favorable for the formation of toroidal pores. A complete set of tested parameters and calculated free energy profiles is provided in *Appendix 1—Figure 1* and *Appendix 1—Figure 2*.

## Effect of peptide flexibility and length on the structure of membrane pores

Using the set of sampled pore configurations from the free energy calculations (*Appendix 1—Figure 1* and *Appendix 1—Figure 2*), we have classified the observed pore structures occurring in the simulations. Since these simulations were performed in the sub-critical regime (where peptides do not open a pore spontaneously), a bias was applied on the lipids in order to observe multiple pore opening and closing events. The bias modifies the Monte Carlo acceptance criteria by adding a penalty to already visited states (in our case pore sizes). Importantly, no bias was applied on the peptides. Therefore, the peptides were free to adsorb into the membrane pore and sample the most favorable configurations. Due to the low hydrophobic content, that is ~50%, the peptides formed various toroidal pores. Summary of the pore morphology for various peptide properties is shown in *Figure 2* and *Figure 3*. Apart from the common toroidal pores with peptides in the transmembrane orientation, we also observed disordered toroidal pores with the majority of peptides adsorbed near the pore rim and oriented parallel with respect to the membrane plane (*Leontiadou et al., 2006*). Interestingly, flexible peptides with a total length exceeding the membrane thickness can adopt Hourglass (*Jung et al., 1994*) or U-shaped (*Santo and Berkowitz, 2012*) structures. In the Hourglass structure, peptides span the membrane. In the U-shape structure, both peptide termini are anchored on the same leaflet and the peptide inserts into the membrane only with its central part.

Furthermore, we performed unbiased simulations to find the relation between peptide parameters and the pore structures. We considered all combinations of the following peptide parameters: (1) peptide length (4, 5, and 7 nm), (2) hydrophobicity (hydrophobic sector in the range of 150° to 350°), and (3) peptide flexibility (rigid or fully flexible).

Barrel-stave pores are formed by rigid membrane-spanning peptides with high hydrophobic content. In our phenomenological model, peptide-peptide interactions are realized solely via hydrophobic interactions. Thus, significant portion of the peptide surface had to be hydrophobic (i.e., sector of at least 270°). Then, each peptide can effectively interact with both lipid tails and neighboring peptides. Moreover, the peptides that formed barrel-stave pores had hydrophilic ends and closely matching hydrophobic length with the membrane thickness.

Compared to barrel-stave pores, formation of toroidal pores was observed over a broader range of peptide properties. Toroidal pores are formed by weakly associated peptides with hydrophobic sector lower than 270°. Peptides with hydrophobic sector higher than 310° either aggregate in solution or insert into the membrane without pore formation.

| | Kink | Length [nm] | | | | | |
|---|---|---|---|---|---|---|---|
| | | 4.0 | | 5.0 | | 7.0 | |
| PSC-AE | rigid | – | TP | TP | DTP | TP | DTP |
| | flexible | – | TP | TP | U | U/HG | U |
| PSC-NE | rigid | – | – | – | TP | TP | DTP |
| | flexible | – | – | – | HG | HG | HG |

**Figure 2.** The effect of peptide properties and kink flexibility on the morphology of membrane pores. Peptides with higher hydrophobic moment (see the Materials and methods) are written in gray. Toroidal pore (TP); Disordered toroidal pore (DTP); Hourglass pore (HG); U-shaped pore (U). PSC-AE and PSC-NE represent peptides with hydrophobic and hydrophilic termini, respectively.

## MARTINI simulations

We investigated the influence of peptides on the stability of various pores using a more detailed coarse-grained MARTINI model (*Monticelli et al., 2008*). This model represents (on average) four heavy atoms as a single bead and it is thus able to capture the peptide sequence-activity relationship. Since MARTINI force field uses a pre-defined secondary-structure of proteins, we have defined the peptides to be either: (1) fully helical or (2) helical with the exception of flexible kink (all the studied sequences and flexible residues are in *Table 1* in the Materials and methods section).

In the beginning, we created a large pore in an equilibrated membrane and then we inserted several peptides according to the hydrophobic mismatch. Then, we have performed a 3 μs long simulation for every system to determine the preferred peptide orientation inside the pore. In this simulation, the pore size was kept constant which enabled the peptides to quickly reorient and assume more preferred configuration. Subsequently, we have prepared one to four distinct starting configurations and performed independent simulations covering all observed peptide conformations in a large pre-formed pore (see *Appendix 1—table 2* and the Materials and Methods section for more details). Moreover, a simulation with a POPC membrane without peptides was performed and used as a reference.

We have performed simulations of Buforin II (*Figure 4*, *Appendix 1—figure 3*, and *Appendix 1—figure 4*), LL-37 (*Appendix 1—figure 5* and *Figure 7*), Candidalysin (*Appendix 1—figure 6*, *Figure 8* and *Appendix 1—Figure 7*), δ-lysin (*Figure 9* and *Appendix 1—figure 8*), and Magainin 2 (*Figure 9* and *Appendix 1—figure 9*). The figures in Appendix show the density of water molecules inside the membrane region over the course of the simulation. The average number of water molecules in each studied system is in *Appendix 1—table 3*.

### Buforin II

Buforin II (*Park et al., 1998*) wild-type (WT) was experimentally shown to translocate across lipid membranes without the formation of membrane pores. The proline kink, located roughly in the middle of the sequence, was identified as a translocation-promoting factor (*Park et al., 2000*). Here, we studied the peptide properties that change the mechanism of action of Buforin II peptide from translocation to pore formation. We have evaluated the effects of: (1) the kink flexibility, (2) single residue substitutions, and (3) lipid composition on the stability of membrane pores.

In agreement with the reported findings (*Park et al., 1998*), we have observed rapid closing of the pre-formed pores in POPC membrane for Buforin II WT. In contrast, Buforin II WT* (WT sequence with a fully helical structure without a kink) and helical variants P11L and P11A stabilized pore structure for the whole duration of the simulation of 50 μs. Another flexible Buforin II variant with a glycine kink, P11G, was also unable to stabilize membrane pores. Then, we have gradually decreased the flexibility of this variant by selecting three, one, and zero residues to form the kink in P11G, P11G̃, and P11G* respectively (see *Table 1*). Simulations are labelled based on the used peptides. *Appendix 1—figure 4* shows that the pore stability can be improved by decreasing the peptide flexibility. *Figure 4* shows representative snapshots from simulations of Buforin II: (A) barrel-

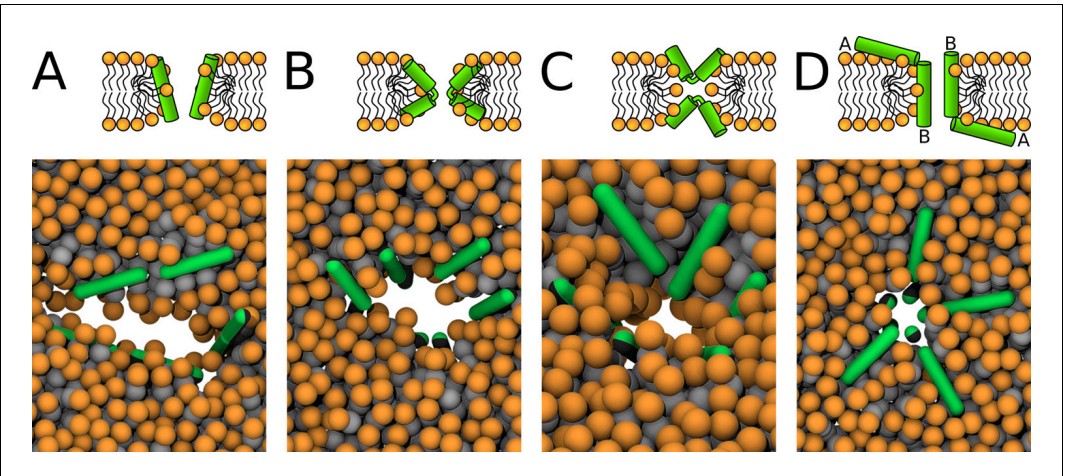

**Figure 3.** Schematic images (top) and representative simulation snapshots (bottom) of various pore structures. (**A**) Toroidal pore with rigid peptides, (**B**) hourglass pore with flexible peptides, (**C**) U-shaped pore, and (**D**) synergistic pore. Schematic images are from a side view, while snapshots are from top view on the membrane. Color coding: lipid headgroup = orange, lipid tail = light gray, hydrophilic = green, hydrophobic = dark gray.
The online version of this article includes the following source data for figure 3:

**Source data 1.** System configurations for snapshots.

stave pores formed by P11L and (B) single partially inserted P11G peptide. The analysis of pore stability for all systems is shown in *Appendix 1—table 3* and *Appendix 1—figure 3*.

The stability of Buforin II-formed barrel-stave pores was similar for systems with all considered lipid compositions (POPC, POPC:POPG (1:1 mol/mol), and POPC:POPS (1:1 mol/mol)), see *Appendix 1—figure 3* and *Appendix 1—table 3*. Generally, the addition of charged lipids reduced the pore diameter and made the peptide packing even more compact, see *Appendix 1—figure 3*. The P11L variant formed the most stable pore across all of the considered membranes. Representative snapshots of the pore structures with P11L are shown in *Figure 5*.

The presence of negatively charged lipids affects the morphology of barrel-stave pores. The negatively charged lipid headgroups of POPG and POPS lipids were preferentially pulled into the pore lumen by the positively charged peptide side-chains. To evaluate this effect, we calculated the

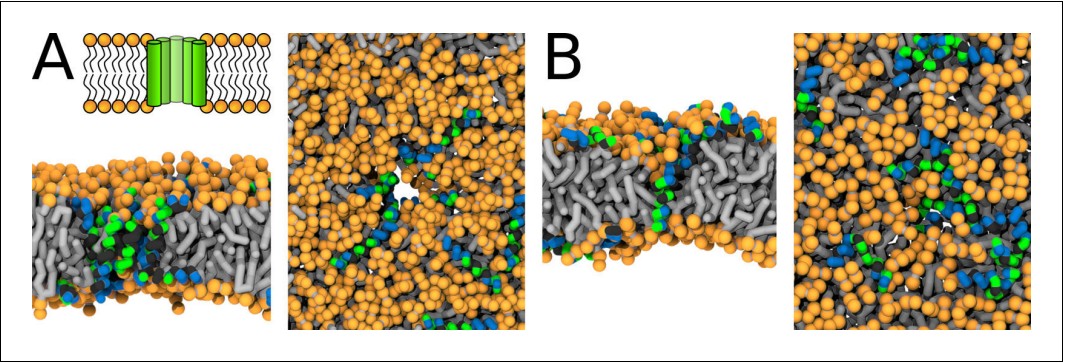

**Figure 4.** Representative snapshots of MARTINI simulation of Buforin II peptide in POPC membrane. Buforin II variants (**A**) P11L (fully α-helical) and (**B**) P11G (helix-kink-helix) are shown. (**A**) Schematic image of barrel-stave pore (top left) and simulation snapshots from side view (left) and top view (right) of the pore are shown. Color coding: hydrophobic residues = dark gray, hydrophilic residues = green, positively charged residues = blue, lipid phosphate group = orange, lipid tail = light gray.
The online version of this article includes the following source data for figure 4:

**Source data 1.** System configurations for snapshots.

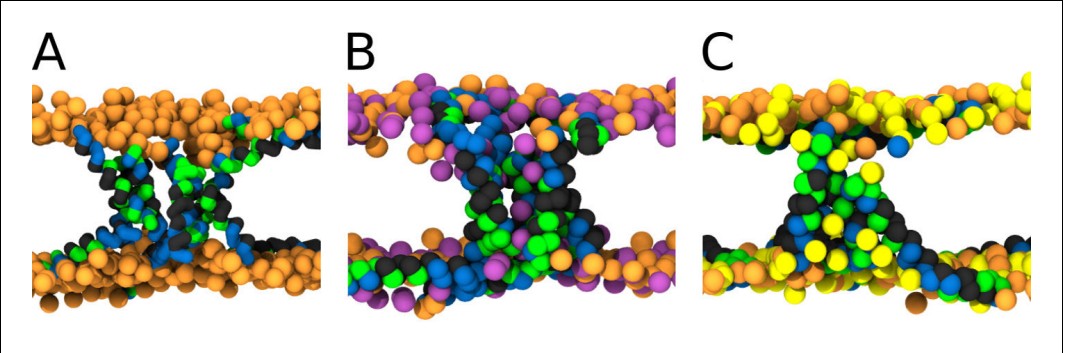

**Figure 5.** MARTINI simulation snapshots of Buforin II P11L peptide. Side views on membranes composed of (**A**) POPC, (**B**) POPC:POPG (1:1 mol/mol), and (**C**) POPC:POPS (1:1 mol/mol) lipids are shown. For clarity, only phosphate groups of lipids are shown. Color coding: hydrophobic residues = dark gray, hydrophilic residues = green, positively charged residues = blue, POPC (orange), POPG (purple), and POPS (yellow).

The online version of this article includes the following source data for figure 5:

**Source data 1.** System configurations for snapshots.

phosphate density profiles within 2.5 nm of the pore center and compared it with the rest of the membrane. Despite the same charge, POPG lipids were buried deeper in the hydrophobic core compared to POPS. For more details, see the phosphate density profiles in *Figure 6*. Furthermore, the lipid density profiles show the effect of the hydrophobic mismatch and the resulting local membrane thinning. Although Buforin II is capable to span the whole lipid bilayer, the hydrophobic residues form a continuous stripe of only ~2 nm long along the peptide long axis.

## LL-37

LL-37 is a highly charged peptide that is a part of human innate immune system (*Turner et al., 1998*). In an ideal α-helical structure, the peptide length would be ~5.5 nm. Thus, the peptide is longer that the typical thickness of the membrane hydrophobic core.

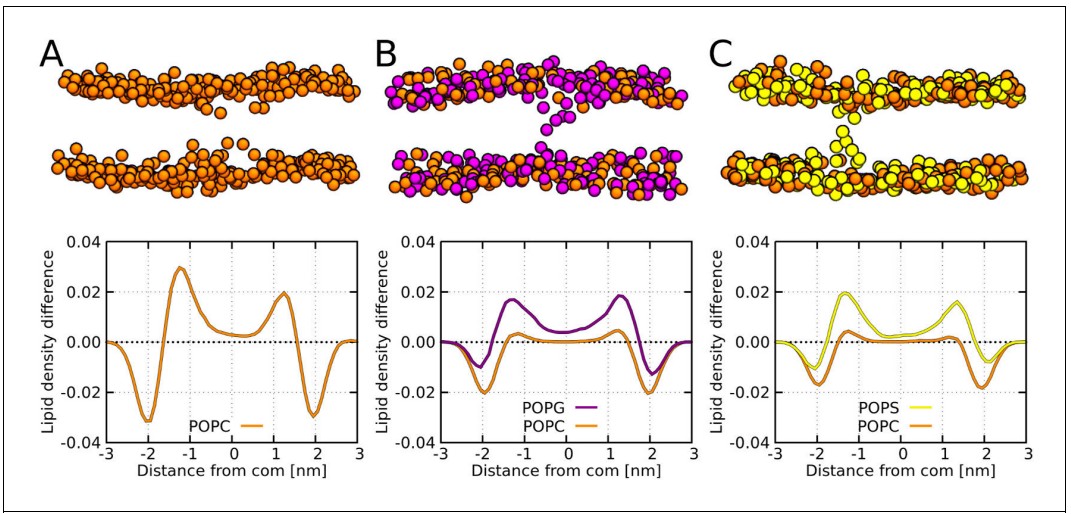

**Figure 6.** MARTINI simulation snapshots and density profiles of Buforin II P11L peptide. Differences between density profiles of phosphates within 2.5 nm of the pore center and the rest of the membrane are shown. Membranes composed of (**A**) POPC, (**B**) POPC:POPG (1:1 mol/mol), and (**C**) POPC:POPS (1:1 mol/mol) lipids are shown. For clarity, only phosphate groups are depicted as spheres. Color coding: POPC (orange), POPG (purple), and POPS (yellow).

The online version of this article includes the following source data for figure 6:

**Source data 1.** System configurations for snapshots and density profiles.

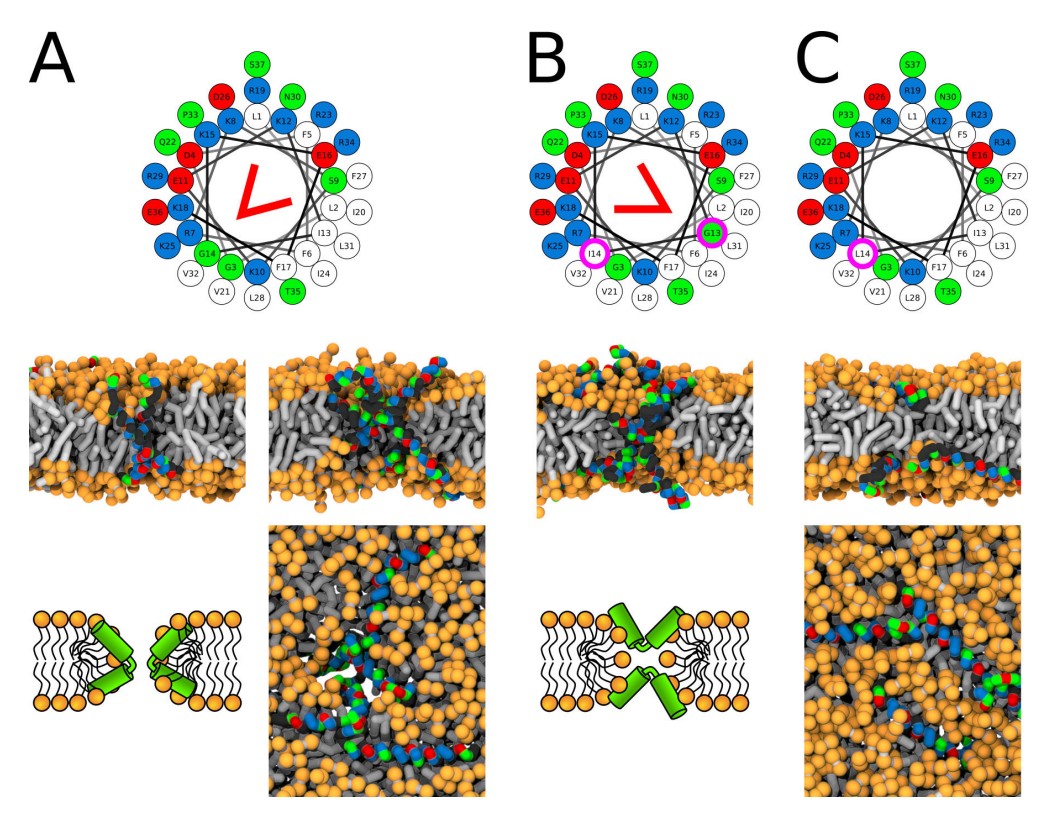

**Figure 7.** Representative snapshots of MARTINI simulation of LL-37 peptide in POPC membrane. LL-37 variants (**A**) WT, (**B**) I13G/G14I, and (**C**) G14L are shown. Coloring of the helical wheel (top) corresponds to Wimley-White octanol scale. Red wedge shows position of the kink. The positions of substituted residues are highlighted with magenta circles. Both side (schematic images and snapshots in middle of the figure) and top views (bottom snapshots) on the membrane structures are shown. Color coding of snapshots: hydrophobic residues = dark gray, hydrophilic residues = green, positively charged residues = blue, negatively charged residues = red, lipid phosphate group = orange, lipid tail = light gray.

The online version of this article includes the following source data for figure 7:

**Source data 1.** System configurations for snapshots.

A kink-induced flexibility of the LL-37 peptide is crucial for the formation of toroidal pores in POPC membrane. The flexible variants (i.e., WT and I13G/G14I) were able to stabilize toroidal pores over the whole duration of the simulation runs. The minimum diameter of the LL-37 WT pore was ~0.5 nm. In contrast, fully-helical variants preferred being adsorbed on membrane surface. The pre-formed pore with LL-37 WT* (fully helical secondary structure without kink) completely closed within 5 μs and the pore with G14L variant formed only transiently. See *Appendix 1—figure 5* and *Figure 7A–C* for more details.

The position of the kink in the LL-37 sequence changes the pore morphology from hourglass to U-shaped. In the LL-37 WT sequence, the the kink-forming glycine residue (*Xhindoli et al., 2016*) is at the interface between hydrophobic and hydrophilic sectors. *Figure 7A* shows the position of the kink and a structure of the so-called hourglass-shaped pore. In the pore, the kink tends to bend the peptide by approximately 90° with the hydrophobic surfaces of the two helical segments facing towards each other. A section through the membrane highlighting the mutual positions of two peptides is also shown. If the glycine residue is in the middle of the hydrophobic sector (I13G/G14I variant), then the peptide can insert into the membrane pore with its central kink, while the termini stay anchored on the same membrane leaflet, see *Figure 7B*. This topology corresponds to an U-shaped pore. The peptide is still bent, but the hydrophobic surfaces are facing away from each other.

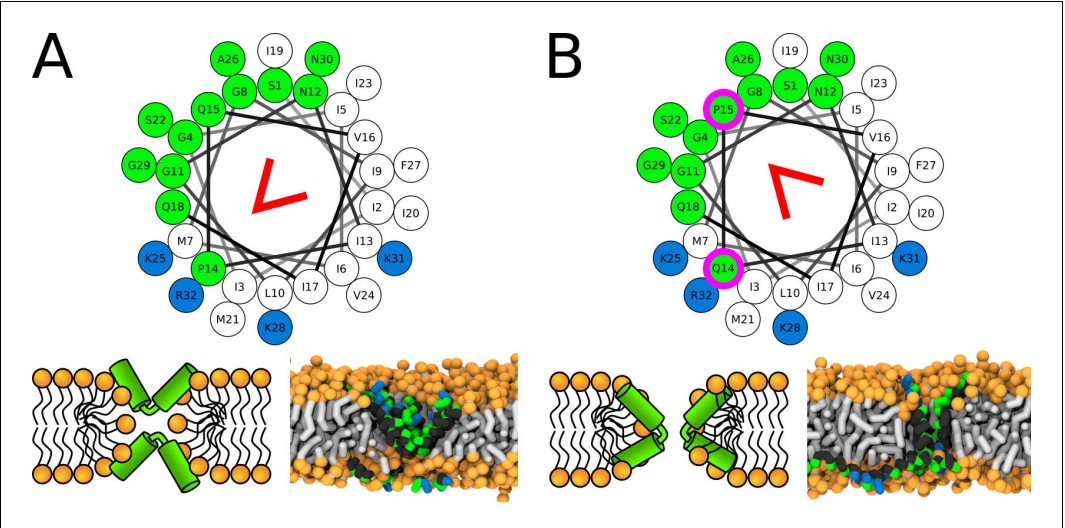

**Figure 8.** Representative snapshots of MARTINI simulation of CandKR peptide in POPC membrane. CandKR variants (**A**) WT and (**B**) P14Q/Q15P are shown. Coloring of the helical wheel (top) corresponds to Wimley-White octanol scale. Red wedge shows position of the kink. The positions of substituted residues are highlighted with magenta circles. Color coding of snapshots: hydrophobic residues = dark gray, hydrophilic residues = green, positively charged residues = blue, lipid phosphate group = orange, lipid tail = light gray.

The online version of this article includes the following source data for figure 8:

**Source data 1.** System configurations for snapshots.

## Candidalysin

Candidalysin is a cytolytic toxin produced by *Candida albicans* (*Moyes et al., 2016*). As part of the maturation process, Candidalysin is enzymatically truncated several times (*Moyes et al., 2016*). We have studied two variants of the peptide: (1) with two positively charged residues at the C-terminus, denoted here as CandKR, and (2) without the terminal arginine, denoted as CandK. The full sequences and the assigned secondary structures are shown in *Table 1*.

Similar to the simulations of the LL-37 peptide, re-positioning of the kink-forming glycine residue changes the morphology of the pore structure. In the CandKR sequence, the kink is at the interface between the hydrophobic and hydrophilic sectors. Unlike the LL-37 WT sequence, the hydrophobic regions are facing away from each other. *Figure 8A* shows a representative snapshot of the U-shaped pore. Placing the kink-forming residue in the middle of the hydrophilic patch (P14Q/Q15P)

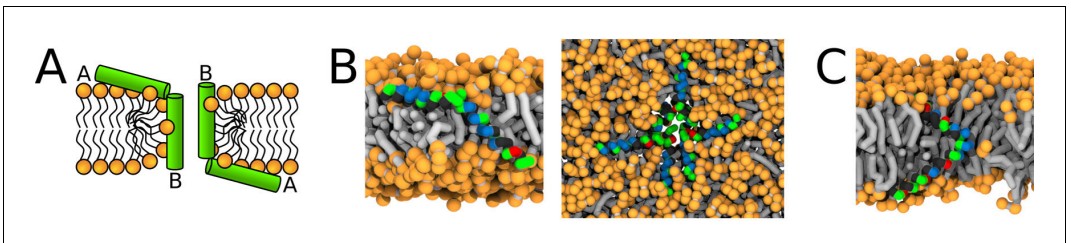

**Figure 9.** Representative snapshots of MARTINI simulation of Magainin 2 and δ-lysin peptides in POPC membrane. (**A**) Schematic image of a synergistic pore structure, (**B**) side (left) and top (right) views on Magainin 2-formed pores, and (**C**) side view on two δ-lysin peptides in a pore. Color coding: hydrophobic residues = dark gray, hydrophilic residues = green, positively charged residues = blue, negatively charged residues = red, lipid phosphate group = orange, lipid tail = light gray.

The online version of this article includes the following source data for figure 9:

**Source data 1.** System configurations for snapshots.

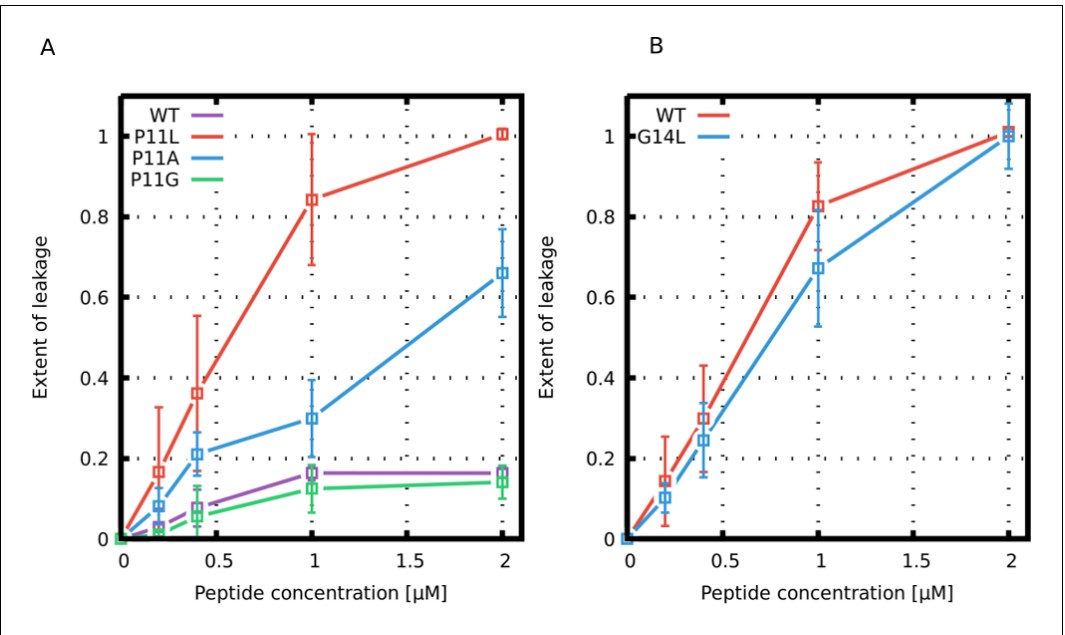

**Figure 10.** Leakage assay with calcein-loaded large unilamellar vesicles. Leakage induced by (**A**) Buforin II and (**B**) LL-37 derived peptides is shown.

The online version of this article includes the following source data for figure 10:

**Source data 1.** Vesicle leakages as a function of peptide concentrations.

results in the formation of a hourglass pore, see *Figure 8B*. In this variant, the hydrophobic regions face towards each other.

During an infection, the predominant form of secreted Candidalysin is terminated with the lysine residue, here denoted as CandK (*Moyes et al., 2016*). Experimentally, this peptide was shown to insert into membranes and increase their permeability (*Moyes et al., 2016*). Our simulations show that the peptide forms only transient toroidal pores, see *Appendix 1—Figure 6*. The exact pore morphology, however, was ambiguous. In different simulations, we have observed the peptides to be oriented both perpendicular and in the U-shaped conformation.

## Magainin 2 and δ-lysin

Magainin 2 is an amphibian AMP forming toroidal pores (*Ludtke et al., 1996*; *Matsuzaki et al., 1998b*; *Yang et al., 2001*). In the initial stage of the pore formation, Magainin 2 was suggested to form a large transient pore that shrinks gradually and becomes more stable (*Tamba et al., 2010*). Together with the induced lipid flip-flop (*Matsuzaki, 1998a*), the data suggests that Magainin 2 forms toroidal pores. Furthermore, atomistic simulations of a more hydrophobic Magainin 2 analogue have shown that majority of the pore-forming peptides are lying in a predominantly parallel orientation close to the pore rim (*Leontiadou et al., 2006*).

δ-lysin is a hemolytic peptide (without antimicrobial activity) expressed by *Staphylococcus aureus* (*Alouf et al., 1989*; *Dhople and Nagaraj, 2005*). Introduction of proline residue in the sequence reduced the hemolytic activity and the replacement of all aspartic acid residues to lysine was shown to induce antimicrobial activity (*Dhople and Nagaraj, 2005*).

Both Magainin 2 WT* and δ-lysin peptides were observed to form synergistic pores in our simulations. In agreement with experimental findings, Magainin 2 with a glycine kink formed a toroidal pore (*Yang et al., 2001*). In contrast, Magainin 2 WT* (fully helical) peptides were associated via N- and C-termini (*Figure 9B*). One peptide was largely in a parallel orientation with the membrane plane while the other was inserted in the pore. The angle between the pair of peptides was ~135°. Similar behavior was observed for δ-lysin peptides (*Figure 9C*). However, δ-lysin peptides associated via C-termini and the interaction took place in the middle of the membrane. Such structure resembles the hourglass pore observed with peptides containing the helix-kink-helix motif. The distribution

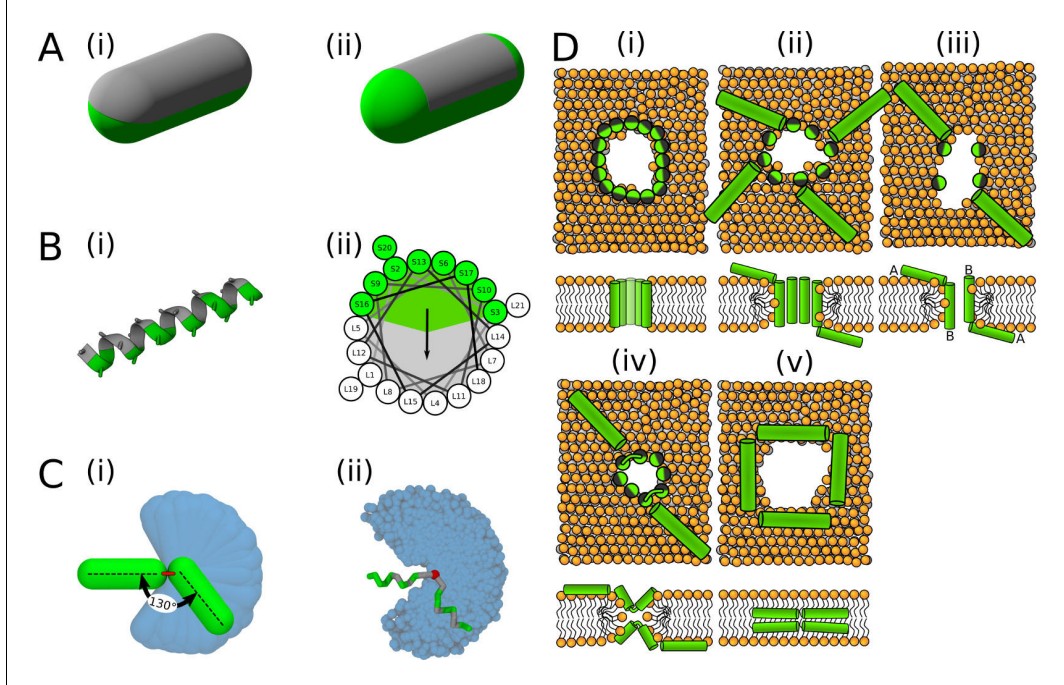

**Figure 11.** Visualisation of the employed models and initial configurations. (**A**) Phenomenological model of half hydrophobic (gray) and half hydrophilic (green) helix peptides. The hydrophobic surface (**i**) extends till the ends in PSC-AE, or (**ii**) is confined to the cylindrical part in PSC-NE. A half-hydrophobic peptide (*Lear et al., 1988*) represented in (**i**) MARTINI and (**ii**) helical wheel projection (hydrophobic sector is shown as a gray area). (**C**) Comparison of the helical kink flexibility using our (**i**) PSC and (**ii**) MARTINI models. The central kink is colored in red. A peptide configuration with kink of roughly 130° is depicted on top of the range of available kink configurations depicted in blue. (**D**) Schematic illustration of the employed initial configurations for simulations with MARTINI model. Each configuration (**i–v**) is shown in the top and side view in upper and lower row, respectively. Green/gray depicts half hydrophilic/hydrophobic peptides. Lipids are displayed with orange headgroups and silver tails. One to four of these distinct pore structures were used for each peptide as the initial configurations.

of angles between the associated peptides during the simulation can be found in *Appendix 1—Figure 10*. These two examples demonstrate the structural similarity of pores formed by peptides with a helix-kink-helix motif and peptides interacting via their termini.

## Calcein leakage assay

Assays with liposomes loaded with fluorescent dyes are standard methods used to evaluate membrane permeability. When the membrane permeability increases, for example after the addition of pore-forming AMPs, a dye is released from the vesicles into surrounding solution. The dye separation from a quencher causes measurable change in fluorescence signal that can be related to the the peptide activity.

Similar to a previous study of the Buforin II peptide (*Kobayashi et al., 2000*), we have prepared large unilamellar vesicles (LUVs) composed of POPC:POPG 1:1 (mol/mol) vesicles and measured the dye leakage at 25°C. *Figure 10* shows our results for Buforin II and LL-37 derived peptides. Substitution of the kink-forming proline residue with leucine (P11L) in Buforin II dramatically increased the membrane permeability. The second most potent Buforin II variant was helical but less hydrophobic P11A variant. In contrast, peptides with a kink (higher flexibility), the wild-type and P11G variant, exhibited a lower leakage of the fluorescent dye. Fluorescence leakage assays of LL-37 peptides show that G14L substitution slightly hindered membrane leakage, which is in agreement with our MARTINI simulations.

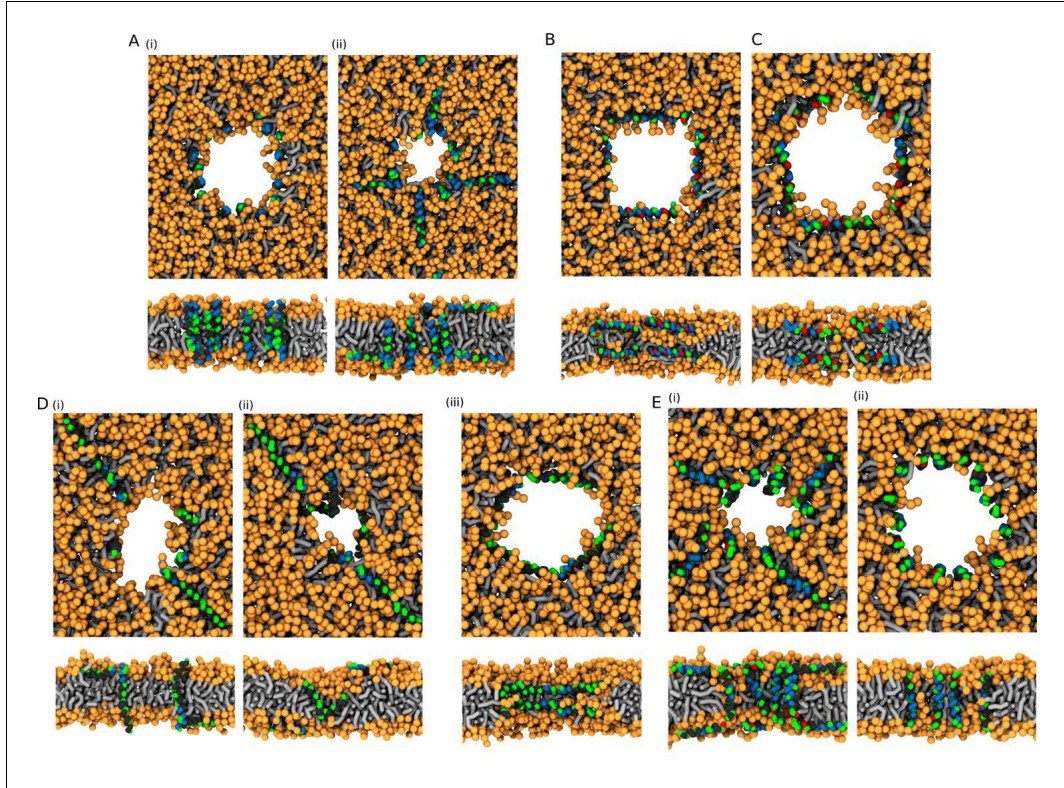

**Figure 12.** MARTINI simulations – initial conditions. Based on the preferred peptide orientation within membrane pore, couple of initial configuration were suggested and tested. (**A**) Buforin II peptides (**i**) 16 peptides oriented perpendicular to membrane plane with antiparallel arrangement in a pre-formed pore, (**ii**) eight peptides oriented perpendicular to membrane plane with antiparallel arrangement in a pre-formed pore and eight peptides being placed on membrane leaflets parallel to membrane, (**B**) 8 LL-37 peptides positioned parallel to membrane plane in pre-formed pore with double-belt pore arrangement, (**C**) eight δ-lysin peptides being positioned parallel to membrane in pre-formed pore with double-belt pore arrangement, (**D**) Candidalysin peptides (**ii**) four peptides placed in a pre-formed pore with perpendicular orientation to the membrane plane and four peptides on membrane surface oriented parallel to it, (**ii**) four peptides in a pre-formed pore in U-shaped conformation and four peptides on membrane, (**iii**) eight peptides in the pore oriented parallel to the membrane plane as in double-belt pore arrangement, (**E**) Magainin 2 peptides (**I**) eight peptides in the pore with antiparallel mutual orientation perpendicular to the membrane plane and eight peptides on membrane with parallel orientation to the membrane plane, and (**ii**) 16 peptides in the pore with antiparallel mutual orientation perpendicular to the membrane plane. Color coding: Hydrophobic residues = black, hydrophilic residues = green, positively charged residues = blue, negatively charged residues = red, lipid heads = orange, lipid tails = light gray.

The online version of this article includes the following source data for figure 12:

**Source data 1.** System configurations for snapshots.

## Discussion

### Energetics of pore formation

Pore formation in a lipid membrane is an energetically unfavorable process (*Kabelka and Vácha, 2015*). The barrier for pore formation can be decreased by peptide adsorption in a concentration-dependent manner (*Huang and Wu, 1991*; *Lee et al., 2004*). Generally, AMPs bind more strongly into the highly-curved pore lumen compared to the membrane surface, thus providing a driving force for pore opening (*Kabelka and Vácha, 2015*). After a critical concentration of AMPs is reached, pore formation becomes spontaneous (*Kabelka and Vácha, 2015*) and observable in experiments. By contrast, computer simulations can be performed in a sub-critical regime, that is below the critical concentration. In such cases, various enhanced sampling methods can be used to facilitate the pore opening and calculate the associated free energy.

In this work, we employed Metropolis MC simulations with the Wang-Landau method (*Wang and Landau, 2001*) for free energy calculations. During the simulation, dozens of pore opening and closing events occur before the free energy profile converges. The results are suitable for assessment of the peptide effectiveness. Note that no bias is applied on the peptides and many possible peptide distributions inside the pore are sampled. The error of these simulations is very low and in the range of a few kT (below kcal/mol).

Due to the high computational cost of the free energy calculations, we use a phenomenological (highly coarse-grained) model to describe secondary amphiphilic peptides, see *Figure 11*. A designed peptide with similar properties (composed of only leucine and serine residues) was experimentally shown to form pores (*Lear et al., 1988*). In a helical wheel representation (*Schiffer and Edmundson, 1967*), the hydrophobic content of this peptide can be described as a circle sector of ~120°. Moreover, we verified the conclusions of the MC simulations with several well-studied peptides using a more detailed MARTINI model (see below).

## Opposite effect of a kink on Barrel-stave and Toroidal Pore formation

Using computer simulations, we identified that presence of a kink in helical peptides destabilizes barrel-stave pores, while peptides forming toroidal pores benefit from kink-induced increased flexibility. Most of our studies were done using a single-component POPC membrane as a general model of a neutral membrane created from a common lipid. Since negatively charged lipids were shown to be important for AMPs, we also tested a mixtures of POPS and POPG lipids, which are negatively charged lipids common in eukaryotic and bacterial cells, respectively. We used the lipid mixtures 1:1 (mol/mol) to be able to capture locally increased concentration of charged lipids in the vicinity of positively charged peptides (*Krauson et al., 2015*) and to prevent the studied behavior from being diffusion limited. Note that some bacteria were reported to contain even higher fraction charged lipids (e.g., 70% in *Bacilus subtilis* [*Clejan et al., 1986*]). Furthermore, lipid composition POPC:POPG 1:1 (mol/mol) was suggested as a suitable membrane model of bacteria based on the correlation of the leakage experiments and the minimum inhibitory concentration (MIC) measurements in live bacteria (*Cheng et al., 2011*).

### Barrel-Stave pores

MC simulations highlighted the three conditions for efficient formation of barrel-stave pores: (1) peptide rigidity, (2) strong peptide-peptide interaction, and (3) peptide length.

Firstly, rigid peptides are more suitable for the formation of barrel-stave pores. The MC simulations with the phenomenological model enabled us to calculate free energy of pore formation for peptides with and without kink (*Figure 1*). The peptide without a kink has markedly lower free energy of pore formation, compared to the peptides with a fully flexible kink. The difference originates in an entropic effect because linear peptides fit better into the geometry of barrel-stave pore and the peptide flexibility hinders the necessary tight packing of the peptides. Furthermore, our more detailed MARTINI simulations showed that α-helical variants (WT*, P11L, and P11A) of Buforin II peptide formed and stabilized the barrel-stave pore structure for the entire duration of the simulations (50 μs). In contrast, Buforin II variants with a kink (i.e., WT [*Park et al., 1998*] and P11G) were not able to stabilize any membrane pores and ended up adsorbed on the membrane surface. Together, these simulations suggest that the peptide rigidity is more important than the chemical nature of the kink-forming residue for the stabilization of barrel-stave pores.

The observed effects of increased peptide rigidity on pore stability is in agreement with previously reported experiments (*Kobayashi et al., 2000*; *Xie et al., 2011*; *Park et al., 2008*; *Liu et al., 2011*; *Jacob et al., 1999*). An enhanced membrane permeabilization by the more helical P11A variant (compared to WT) of Buforin II has already been experimentally demonstrated (*Kobayashi et al., 2000*). Moreover, flexible variants P11A/V12P and P11A/V15P of Buforin II had lower membrane permeabilization activity compared to P11A variant (*Xie et al., 2011*). HP(2–20) was suggested to form narrow channels (*Park et al., 2008*) and its antimicrobial activity is rigidity-enhanced (*Liu et al., 2011*). However, increased rigidity (P14A/G11A variant) of barrel-stave pore forming Alamethicin did not significantly alter the pore characteristics (*Jacob et al., 1999*).

Secondly, strong peptide-peptide interactions are needed to stabilize the bundle-like assembly of peptides. Specifically, the hydrophobic sector of the peptides in our MC simulations had to be at

least 270° in order to mediate both peptide-peptide and peptide-lipid interactions, which is in line with previous simulations (*Illya and Deserno, 2008*) and experiments on Alamethicin (*Yang et al., 2001*).

Buforin II P11L formed the most stable barrel-stave pores in our MARTINI simulations. Thus, we used this variant to study the effect of lipids. The overall stability of the pore structures was comparable for all considered lipid compositions (POPC, POPC:POPG (1:1 mol/mol), and POPC:POPS (1:1 mol/mol)). However, the presence of negatively charged lipids caused a deviation from the regular barrel-stave pore topology. The origin of this effect is probably in electrostatic attraction between positively charged side-chains of Buforin II (net charge +6) and the negatively charged headgroups of POPG and POPS lipids. This interaction caused tilting of the charged lipid headgroups towards the membrane core and destabilization of the barrel-stave pore structure.

Lastly, the length of barrel-stave-pore-forming peptides should match the membrane thickness. For the formation of a barrel-stave pore, the peptide should be oriented roughly perpendicular with respect to the membrane plane. Thus, the peptide should comply with the hydrophobic mismatch. Longer peptides did not form barrel-stave pores. The effect of hydrophobic mismatch on pore forming peptides has been demonstrated both theoretically (*Vácha and Frenkel, 2013*) and experimentally (*Grau-Campistany et al., 2015*). Hydrophilic or charged residues at the peptide termini further stabilize perpendicular orientation via interactions with charged moieties of lipid heads (*Vácha and Frenkel, 2013*).

## Toroidal pores

Peptides inside a toroidal pore are not regularly packed and both peptides and lipids are present in the pore lumen. Our simulations revealed several characteristics of peptides in toroidal pores: (1) self-association is not essential, (2) increased flexibility generally improves activity, (3) position of the kink can alter the pore structure, and (4) surface-aligned peptides (outside the pore) can still participate in pore stabilization.

Toroidal pores were observed over a broader range of peptide properties compared to the peptides forming barrel-stave pores. In many cases, peptide self-association was not very pronounced and hydrophobic content of ~50% was found to be sufficient for the formation of toroidal pores. Hydrophobic sector higher than ~310° suppressed the pore-forming activity, as the peptides aggregated in solution or interacted with the membrane (adsorption or insertion) without pore formation.

Increased peptide flexibility has a positive effect on the stability of toroidal pores. Flexible peptides can more easily adapt to the curved (catenoid) shape of the pore lumen and our MC simulations (see *Figure 1*) show a decrease in a free energy of pore formation after the introduction of a kink. Furthermore, stronger partitioning of peptides with an imperfect α-helical secondary structure into the membrane core was observed experimentally (*Henderson et al., 2016*). Nevertheless, including a flexible proline/glycine kink into a real peptide sequence would also alter the peptide affinity for membrane due to modified hydrophilicity. Indeed, the reduced adsorption was demonstrated for Melittin L16G (a variant with increased conformational flexibility) (*Krauson et al., 2015*). After compensating for the decreased adsorption by the addition of negatively charged lipids, the variant regained its permeabilization ability (*Krauson et al., 2015*). In this work, however, we only focused on the effect of peptide flexibility. Thus, we took advantage of features of the MARTINI model and used multiple definitions of the peptide secondary structure to circumvented this problem altogether. We used the same peptide sequences and tuned the flexibility of the peptides (these simulations are marked with an asterisk). Nonetheless, the change in peptide adsorption could exceed the effect on the pore stabilization and simple addition of a kink-forming residue may not be always a viable strategy.

Our MARTINI simulations verified the importance of peptide flexibility predicted by our MC simulations with phenomenological model. LL-37 peptide has a kink-forming glycine at position 14 (*Xhindoli et al., 2016*) and disrupts POPC membranes via the formation of toroidal pores (*Lee et al., 2011*). Indeed, the WT peptide stabilized the toroidal pore for the entire duration of the simulation. When the peptide was kept in a α-helical conformation (WT*), the stability of the toroidal pore was decreased and pore closing was observed after 10 µs. Similarly, lower pore stability was observed for a more hydrophobic α-helical LL-37 G14L variant. In an another example, flexible

Candidalysin (CandK) peptides were able to stabilize small toroidal pores. When the proline residue at position 14 was restrained in a fully helical conformation (WT*), the pre-formed pores closed.

AMPs with the helix-kink-helix motif are able to form structurally diverse toroidal pores. Here we focus on two distinct toroidal pore structures with different arrangements of peptides that were described previously: (1) U-shaped (*Santo and Berkowitz, 2012*) and (2) hourglass (*Jung et al., 1994*) pores. The first structure was named after the shape of the peptides which adopt a bent conformation. Each peptide stays on one leaflet with both termini being anchored in the lipid head-group region, while the peptide kink is inserted deeply in the membrane. In the hourglass pore model, peptides span the membrane and the kink is positioned in the membrane center. Both of these pores were observed in our MC (*Figure 3B–C*) and MARTINI simulations.

Position of the proline/glycine kink determines the type/structure of toroidal pores. LL-37 peptide was experimentally shown to form toroidal pores with peptides in a perpendicular orientation with respect to membrane plane (*Lee et al., 2011*). In agreement with these findings, we observed LL-37 to form hourglass-shaped toroidal pores. The kink-forming glycine residue in LL-37 peptides is located on the side of the hydrophilic region (see *Figure 7A*). By rearranging two adjacent residues (LL-37 I13G/G14I variant), the kink-forming residue was moved to the center of hydrophobic sector. In our MARTINI simulations, this change resulted in a formation of a U-shaped toroidal pores. In contrast to LL-37, wild type Candidalysin peptides have the proline residue on the side of the hydrophobic sector (see *Figure 8A*). CandK and CandKR WT peptides were observed to adopt 'U-shaped' (bent) structure inside pores in our simulations. Repositioning the kink residue (CandidKR P14Q/Q15P variant) to the center of hydrophilic sector caused the formation of hourglass-shaped toroidal pores with transmembrane peptide orientation. Therefore, the position of the kink-forming residue has been identified as a key structural determinant which can govern the formation of distinct toroidal pores.

Additional type of toroidal pore was recently suggested to be formed by a pair of associated peptides (*Salnikov and Bechinger, 2011*; *Strandberg et al., 2016*). In this synergistic pore, Magainin 2 peptide was reported to be inserted in the pore and form a polar channel, while the other peptide, PGLa, remained adsorbed on the membrane surface in a parallel orientation with respect to the membrane plane (*Salnikov and Bechinger, 2011*; *Strandberg et al., 2016*). The peptides are strongly interacting via their C-termini forming a stable flexibly connected pair, which could be analogous to the helix-kink-helix motif within a single peptide. Note that strong self-association via the peptide C-termini is possible even for two highly charged polyarginine peptides (*Tesei et al., 2017*). Indeed, we observed synergistic pores with our phenomenological model for long helices with a flexible kink (i.e. connection), see *Figure 3D*. We found that the inner angle of 135° is more suitable for pore formation compared to an angle of 90°. Similar peptide configuration was observed experimentally for Magainin 2 and PGLa peptides (*Strandberg et al., 2016*).

The similarity of pores formed by peptides with the helix-kink-helix motif and by peptides associating via termini is more general and includes self-associating peptides. Magainin 2 peptides were reported to form homomeric pores (*Wakamatsu et al., 2002*). In our MARTINI simulations, surface-aligned and transmembrane Magainin 2 peptides cooperatively stabilized a toroidal pore in a structure similar to the synergistic pore of Magainin 2 and PGLa. The peptide pairs interacted via N- and C-termini and the angle between peptides was approximately 135° (*Figure 9B*). Additional example of toroidal pore formed by self-associating peptides was observed for δ-lysin peptides (*Figure 9C*). δ-lysin peptides interacted via C-termini and the position of the contact between the peptides was in the middle of the membrane. The pore structure of δ-lysin resembled the hourglass toroidal pore, adopted by peptides with the helix-kink-helix motif. These examples demonstrate the analogy between long flexible peptides and peptides interacting via their termini.

## Fluorescent leakage assays of buforin II, LL-37, and their variants

Our simulations with both the phenomenological and MARTINI model revealed that peptide flexibility modulates the formation of barrel-stave and toroidal pores. In order to experimentally validate our computational findings, Buforin II and LL-37 peptides and their variants were synthesized and tested using fluorescence leakage assays with LUVs (for details, see the Materials and methods section). For Buforin II peptides, the largest leakage was observed for P11L, followed by a slightly less hydrophobic P11A variant (see *Figure 10*). Significantly lower leakage was found for Buforin II WT with a proline kink, and the lowest leakage was caused by P11G. This is in excellent agreement with

the stability of pores in our simulations (see *Figure 4* and *Figure 7*) and previous experimental findings (*Kobayashi et al., 2000*; *Park et al., 1998*). Moreover, LL-37 WT peptide exhibited greater leakage than the G14L variant, also in agreement with the simulations. Together with the simulation data, the results show that the presence of a flexible kink hinders the formation of barrel-stave but promotes toroidal pores.

## Conclusions

We investigated the effect of proline/glycine-induced kink in helical amphiphilic peptides on their ability to form leaky pores in model phospholid membranes. Two independent models (highly-coarse grained phenomenological model and coarse-grained MARTINI model) and fluorescence leakage experiments consistently showed that flexible kink in helical peptides facilitated the formation of toroidal pores but destabilized barrel-stave pores.

In particular, we found that fully helical variants (P11A and P11L) of Buforin II stabilized barrel pores, while the wild-type sequence and P11G variant with a central kink did not. The formation of toroidal pores was investigated with LL-37, Candidalysin, δ-lysin, and Magainin 2 peptides and their variants. The exact structure of toroidal pores (Hourglass or U-shaped) was modulated by the position of the kink in the sequence. Moreover, we showed that a flexible connection between two helices could have the same effect as a kink within the structure of a single peptide. In other words, toroidal pores can also be stabilized by association between surface-aligned and transmembrane peptides that strongly interact via termini, effectively forming a helix-kink-helix motif. We observed such pore structures Magainin 2 and δ-lysin peptides.

This study provides a comprehensive molecular rationalization of the effect of a proline/glycine kink in helical peptides in the context of membrane pore formation. As a rule of thumb, most pore-forming peptides could benefit from the inclusion of proline/glycine residue in the middle of the sequence (if the decreased peptide-membrane interaction is accounted for [*Krauson et al., 2015*]). Alternatively, self-association between peptide termini can form a kink-like connection with a similar effect. We observed majority of the peptides to form toroidal pores and only highly hydrophobic or strongly self-associating peptides to form barrel-stave pores. The obtained molecular understanding could be used for the design of more potent antimicrobial peptides or less toxic drug-carrying peptides.

## Materials and methods

### Computational simulations

In this study, two conceptually distinct models with a different level of molecular detail were employed. A phenomenological model was used in Monte Carlo simulations and molecular dynamics simulations were performed with the well-established MARTINI model.

#### Metropolis Monte Carlo simulations

Monte Carlo simulations with Metropolis sampling scheme were performed using an in-house software SC (freely available at https://github.com/robertvacha/SC; *Sukeník et al., 2020*; copy archived at https://github.com/elifesciences-publications/SC).

#### Peptide model

A phenomenological model (*Vácha and Frenkel, 2011*) of secondary amphiphilic peptides (*Kaiser and Kézdy, 1983*) was used to model antimicrobial peptides. Each α-helical segment was represented by a single particle, so-called patchy spherocylinder (PSC), with the hydrophobic content defined as a sector in helical wheel projection (*Schiffer and Edmundson, 1967*), see *Figure 11A–B* for details. The diameter of each particle was 1 nm to roughly match the size of an ideal α-helix and the particle length is defined by the length of the cylinder. Two types of particles are distinguished: (1) with attractive end-caps (PSC-AE) and (2) with non-attractive end-caps (PSC-NE). The PSC-AE and PSC-NE particles are used to describe peptides with hydrophobic/capped (e. g., acetylation and amidation) or hydrophilic/charged termini, respectively. Furtermore for peptides with insufficient membrane attraction, we also simulated a version with higher hydrophobic moment (interaction strength per helical segment increased by 1 ε, see the original paper for more

information about the parameterization [*Vácha and Frenkel, 2011*]). These results are shown in the *Appendix 1—Figure 1* and *Appendix 1—Figure 2*.

Peptide with an α-helical kink was modelled by two PSC particles connected by a harmonic bond. Equilibrium length of the harmonic bond was equal to either: (1) particle diameter (i.e., 1 nm) or (2) zero. These two models are labelled as PSC and O-PSC, respectively. In terms of the kink flexibility, we have considered the two limit cases, that is rigid and fully flexible. The peptides with flexible kink can freely assume different peptide conformations. *Figure 11* shows the range of the kink conformational flexibility. Note that the peptide model was developed to be compatible with the lipid model by *Cooke and Deserno (2005)*. In these implicit solvent models, the hydrophilic interaction is repulsive and represented by Weeks-Chandler-Andersen (shifted and truncated Lennard-Jones) potential. The hydrophobic interactions are attractive and the range of the interaction has a $cos^2$ dependence. More details could be found in the original paper or later model applications. (*Vácha and Frenkel, 2011*; *Vácha and Frenkel, 2013*; *Vácha and Frenkel, 2014*; *Kabelka and Vácha, 2015*; *Leber et al., 2018*; *Kabelka and Vácha, 2018*.

### Lipid model

The membrane was described using the Cooke and Deserno implicit-solvent lipid model (*Cooke and Deserno, 2005*). Each lipid was represented by three beads: single hydrophilic bead to describe the lipid headgroup and two hydrophobic beads describing the lipid tails. This model was demonstrated to capture the membrane elastic properties and phase transition of a lipid bilayer (*Cooke and Deserno, 2005*).

### Simulation parameters

Prismatic unit cell of about $17 \times 17 \times 30$ nm with periodic boundary conditions (PBC) was used. The lipid bilayer was assembled in the XY-plane using 500 lipid molecules and kept at zero tension. After membrane equilibration, peptides were added to the system in a random spatial and orientational distribution. The concentration of peptides and lipids was expressed as the Peptide-to-Lipid (P/L) ratio. We simulated systems with P/L 1/250, 1/100, 1/50, and 1/25. Peptide parameters (length, hydrophobicity, and flexibility of kink) were systematically varied to mimic different peptide physical-chemical properties.

### Free energy calculations

Free energy of pore formation was calculated using the Wang-Landau (WL) method (*Wang and Landau, 2001*). The positions of the lipid tails were projected along the membrane normal on a two dimensional grid (bin size of 0.09 $nm^2$). Then, the pore was defined as the largest continuous area (considering PBC) on the grid without lipid tails. The area of the largest pore was selected as a reaction coordinate, which was previously shown to be suitable for the free energy calculations of pore formation (*Kabelka and Vácha, 2015*; *Wang and Frenkel, 2005*). The free energy profiles for small pores with an area less than 2 $mn^2$ were obtained using the Boltzmann inversion of spontaneous pore distribution. Note that no bias is applied on the peptides.

The WL method repeatedly samples the whole range of the reaction coordinate by discouraging the already visited states. Thus, dozens of pore opening and closing events occur and the peptides can sample many distinct pore configurations. Initially, the modification factor, $f$, (*Wang and Landau, 2001*) was set to $10^{-3}$. The calculation of free energy was performed until the $f$ was lower than the pre-defined value of ~$10^{-8}$. In order to increase the precision of sampling, additional simulations were run without further addition of any bias. The calculated histogram of visited states represented the WL sampling error and was used to further improve the free energy profiles.

### Molecular Dynamics simulations

The molecular dynamics simulations were performed using the software package Gromacs 5.0.5 (*Páll et al., 2015*; *Abraham et al., 2015*). MARTINI 2.1 coarse-grained model (*Monticelli et al., 2008*) was used for the description of the whole system. This model is well-established for protein-lipid simulations and it is frequently used to study antimicrobial peptides (*Santo et al., 2013*; *Kirsch and Böckmann, 2016*; *Guha et al., 2019*).

## Peptide preparation

In MARTINI simulations, the protein secondary structure has to be defined *a priori* and is kept throughout the simulations. We used fully α-helical structure of peptides unless the sequence contained a proline or glycine residue. In such cases, we used two secondary structures: (1) α-helical with flexible kink (unstructured, modelled as a random coil) and (2) fully α-helical for comparison. All considered sequences and secondary structures are shown in *Table 1*. Initially, all-atom structures of the peptides were constructed as α-helices using Modeller 9.11. Then, the peptides were converted to coarse-grained representation by `martinize.py` script, provided by the authors of MARTINI force field (available at https://github.com/cgmartini/martinize.py). The backbone angles and dihedrals of fully helical peptides with a proline residue were adjusted to match the parameters of other residues.

## System preparation

The membrane was composed of 500 1-palmitoyl-2-oleoyl-sn-glycero-3-phosphocholine (POPC) lipids and assembled using CHARMM-GUI (*Jo et al., 2008*) in the XY-plane. For comparison with experiments and to determine the effect of lipid composition, we also investigated lipid mixtures with negatively charged 1-palmitoyl-2-oleoyl-sn-glycero-3-phospho-L-serine (POPS) and 1-palmitoyl-2-oleoyl-sn-glycero-3-phosphoglycerol (POPG) lipids. The lipid ratios were POPC:POPG (1:1 mol/mol) and POPC:POPS (1:1 mol/mol). Lipids were distributed equally among both leaflets.

**Table 1.** Overview of peptides used in MARTINI simulations and observed pore structures.

| Peptide | Variant | Sequence and secondary structure | Pore† |
|---|---|---|---|
| LL-37 | WT | **LLG**DFFRKSK EKI**G**KEFKRI VQRIKDFLRN LV**PRTES** | HG |
| | WT* | **LLG**DFFRKSK EKIGKEFKRI VQRIKDFLRN LV**PRTES** | – |
| | G14L | **LLG**DFFRKSK EKI<u>L</u>KEFKRI VQRIKDFLRN LV**PRTES** | – |
| | I13G/G14I | LLGDFFRKSK EK<u>G</u>IKEFKRIVQRIKDFLRN LV**PRTES** | U |
| Buforin II | WT | TRSSRAGLQ**F PV**GRVHRLLR K | – |
| | WT* | TRSSRAGLQF PVGRVHRLLR K | BP |
| | P11L | TRSSRAGLQF <u>L</u>VGRVHRLLR K | BP |
| | P11A | TRSSRAGLQF <u>A</u>VGRVHRLLR K | BP |
| | P11G | TRSSRAGLQ**F** <u>**G**</u>VGRVHRLLR K | – |
| | P11G* | TRSSRAGLQF <u>G</u>VGRVHRLLR K | BP |
| | P11Gˆ | TRSSRAGLQF <u>**G**</u>VGRVHRLLR K | BP/TP |
| Magainin 2 | WT | GIGKFLHSAK K**FGK**AFVGEI MNS | TP |
| | WT* | GIGKFLHSAK KFGKAFVGEI MNS | SP |
| | G13P | GIGKFLHSAK K**F**<u>**P**</u>**K**AFVGEI MNS | TP |
| | G13P* | GIGKFLHSAK KF<u>P</u>KAFVGEI MNS | SP |
| δ-lysin | WT | MAQDIIST**IG D**LVKWIIDTV NKFTKK | SP |
| | WT* | MAQDIISTIG DLVKWIIDTV NKFTKK | TP |
| CandK | WT | SIIGIIMGIL GNI**P**QVIQII MSIVKAFKGN K | U |
| | WT* | SIIGIIMGIL GNIPQVIQII MSIVKAFKGN K | TP |
| CandKR | WT | SIIGIIMGIL GNI**P**QVIQII MSIVKAFKGN KR | U |
| | P14Q/Q15P | SIIGIIMGIL GNIQ<u>**P**</u>VIQII MSIVKAFKGN KR | HG |

Sequence color coding: α-helix = roman, unstructured = bold. Amino acid substitutions in the peptide sequences are underlined.

* a sequence where glycine/proline (usually a kink-forming residues) was forced into an α-helix.

ˆ a sequence where only single residue was defined to be unstructured.

†Toroidal pore (TP); Disordered toroidal pore (DTP); Barrel-stave pore (BP); Hourglass pore (HG); U-shaped pore (U); Synergistic Pore (SP); No pore formation (–).

The peptide pore-forming activity was evaluated by its ability to stabilize pre-formed pores. Membrane pore was created by removing lipids in a cylinder of defined radius from the box center. Then, the peptides were placed into the pore of fixed size (maintained by zero box compressibility in the XY-plane). The large initial pore diameter enabled the peptides to easily change orientation and assume more preferred configuration. During the subsequent simulation, the pore shrinks. The membrane with peptides was solvated with water and NaCl ions at 100 mM concentration. Excess of ions was added to the neutralize the system net charge. Positional restraints with a force constant of 1000 kJ mol$^{-1}$ nm$^{-2}$ were applied on the peptide backbone beads. The system was energy minimized using the steepest descent algorithm and equilibrated in five steps with increasing simulation time step. The unrestrained system was subsequently simulated for 3 μs to obtain the preferred peptide configurations. Based on the peptide arrangement, up to four distinct starting configurations (see *Appendix 1—table 2*) were prepared (see *Figure 11* and *Figure 12*). The majority of the simulated peptides adopted one particular arrangement in a pore, regardless of the initial configuration used. The only exception were the Candidalysin peptides as they were unable to reorient from a trans-membrane starting configuration to an U-shaped conformation. After further equilibration, NpT production dynamics simulations of a minimum length of 20 μs were performed. Pore stability was evaluated using several critera: (1) observed configurations in the trajectory, (2) amount of water beads in the hydrophobic core, and (3) the presence of a continuous water channel.

## Simulation settings

System temperature was kept at 323 K using velocity rescaling algorithm, modified with a stochastic term (*Bussi et al., 2007*). Initial particle velocities were generated according to Maxwell distribution corresponding to 323 K. For equilibration, pressure was maintained via Berendsen barostat at 1.0 bar with coupling constant of 5 ps. For production dynamics, Parrinello-Rahman barostat (*Parrinello and Rahman, 1981*) with coupling constant 12 ps was used. Semiisotropic coupling scheme was employed with compressibility set to $3.10^{-4}$ bar$^{-1}$ in all directions. Leap-frog algorithm for integrating Newton's equations of motion was used with time step of 30 fs. Verlet cutoff scheme was employed with radius of 1.1 nm. Cutoff for both van der Waals and Coulomb interactions was set to 1.1 nm. The relative dielectric constant was set to 15.

## Experiments

### Chemicals

Phospholipids 1-palmitoyl-2-oleoyl-sn-glycero-3-phosphocholine (POPC) and 1-palmitoyl-2-oleoyl-sn-glycero-3-phospho-(1'-rac-glycerol) sodium salt (POPG) were obtained from Avanati Lipids, Inc (Alabaster, AL, USA). Both phospholipids were dissolved in chloroform by the manufacturer. Lipid solutions were stored at −20°C before use. Synthesis and ion exchange of all peptides was done by JPT Peptide Technologies GmbH (Berlin, Germany). Sequences of Buforin II and LL-37 peptides and all tested variants are provided in *Table 2*. Without further purification, peptides were dissolved in phosphate-buffered saline (PBS). The buffer composition was 25 mM NaPi (NaH$_2$PO$_4$ and Na$_2$HPO$_4$ in ratio 3:7), 100 mM NaCl, 1 mM EDTA. The pH was adjusted to physiological value of 7.4. NaH$_2$PO$_4$ · H$_2$O, NaOH, NaCl were obtained from Merck (Darmstadt, Germany). Non-ionic surfactant Triton X-100 and fluorescent dye Calcein were obtained from Sigma-Aldrich (St.Louis, MO USA). Chelating agent EDTA, NaH$_2$PO$_4$ · 7H$_2$O, Chloroform Spectronorm, Methanol technical were

**Table 2.** Overview of peptides used in the fluorescence experiments.

| Peptide | Variant | Sequence |
|---|---|---|
| Buforin II | WT | TRSSRAGLQF**P**VGRVHRLLRK |
| | P11L | TRSSRAGLQF**L**VGRVHRLLRK |
| | P11A | TRSSRAGLQF**A**VGRVHRLLRK |
| | P11G | TRSSRAGLQF**G**VGRVHRLLRK |
| LL-37 | WT | LLGDFFRKSKEKI**G**KEFKRIVQRIKDFLRNLVPRTES |
| | G14L | LLGDFFRKSKEKI**L**KEFKRIVQRIKDFLRNLVPRTES |

Positions of the substituted residues are highlighted in bold.

purchased from VWR (Solon, OH USA). The far-red fluorescent, lipophilic carbocyanine DiIC18(5) oil (DiD) was purchased from Life Technologies Corporation (Eugene, Oregon USA). DiD was dissolved in chloroform and stored at −20°C.

## Leakage assay

We performed leakage assays to study the pore-forming activity of AMPs and their variants. We prepared large unilamellar vesicles (LUV) loaded with concentrated solution of self-quenching fluorescent dye, Calcein. Pore-forming peptides can substantially increase the membrane permeability and cause leakage of the dye load. Due to the dilution of calcein solution, an increase in fluorescence signal can be measured (*Zhang et al., 2001*; *Hamann, 2002*).

For the leakage assays, fresh vesicle suspensions of defined composition were prepared. The lipids dissolved in chloroform were mixed at the desired ratio of POPC:POPG (1:1 mol/mol). DiD was added into the phospholipid mixture at ratio 1:500 (DiD:lipid mol/mol) for fluorescent determination of the lipid concentration. Thin lipid film was created by evaporation of chloroform inside a fume hood. The remaining traces of chloroform were removed in a desiccator overnight or for at least 2.5 hr. Subsequently, the lipid films were hydrated with Calcein buffer (self-quenching 35 mM Calcein, 25 mM NaPi, 20 mM NaCl, 1 mM EDTA, pH = 7.4) and vortexed vigorously to bring all lipids in suspension. Such solution contains multilamellar lipid vesicles (MLVs) (*Arias, 2016*). Therefore, we performed five freeze-thaw cycles at temperatures above the gel-liquid crystalline phase transition (−78.5 °C/2 min and 30 °C/0.5 min) to reduce the vesicle lamelaritty (*Nayar et al., 1989*). LUVs were extruded 50× through 100 nm polycarbonate filter membrane to obtain uniformly sized vesicle (*Mayer et al., 1985*). Untrapped Calcein was removed on HiTrap Desalting Columns 5 × 5 ml (Sephadex G-25 Superfine matrix, cross-linked dextran) by washing with PBS. The concentration of the lipid suspension was then adjusted to 0.02 mM. Fluorescence measurements were performed with a HORIBA Scientific Jobin Yvon FluoroLog-3 Modular Spectrofluorometer (New Jersey NJ USA), interfaced to a computer using FluorEssence V3.8. Excitation and emission wavelengths were set to 495 nm and 520 nm, respectively. All measurements were performed under constant stirring at 25°C, well above the phase transition temperature of the lipid mixture. Peptides dissolved in PBS (0.1 mM) were added directly to the cuvette to obtain the desired Peptide-to-Lipid (P/L) ratio. Finally, 50 µl of nonionic surfactant Triton X-100 was added to lyse all remaining LUVs and determine the maximum fluorescence intensity for normalization.

## Acknowledgements

AT, IK, TK, LS, and RV were supported by the Czech Science Foundation (grant GA17–11571S) and the CEITEC 2020 (LQ1601) project with financial contribution made by the Ministry of Education, Youths and Sports of the Czech Republic within special support paid from the National Programme for Sustainability II funds. SP and MH acknowledge support by the Czech Science Foundation (via 19-26854X). Computational resources were provided by the CESNET LM2015042 and the CERIT Scientific Cloud LM2015085, provided under the programme 'Projects of Large Research, Development, and Innovations Infrastructures'. This work was supported by the Ministry of Education, Youth and Sports from the Large Infrastructures for Research, Experimental Development and Innovations project 'IT4Innovations National Supercomputing Center – LM2015070'.

## Additional information

### Funding

| Funder | Grant reference number | Author |
| --- | --- | --- |
| Czech Science Foundation | GA17-11571S | Alzbeta Tuerkova<br>Ivo Kabelka<br>Tereza Králová<br>Lukáš Sukeník<br>Robert Vácha |
| Czech Science Foundation | 19-26854X | Šárka Pokorná<br>Martin Hof |

| | | |
|---|---|---|
| Ministry of Education, Youth and Sports of the Czech Republic | LQ1601 | Alžběta Türková<br>Ivo Kabelka<br>Tereza Králová<br>Lukáš Sukeník<br>Robert Vácha |
| Ministry of Education, Youth and Sports of the Czech Republic | LM2015070 | Alžběta Türková<br>Ivo Kabelka<br>Tereza Králová<br>Lukáš Sukeník<br>Robert Vácha |
| Ministry of Education, Youth and Sports of the Czech Republic | LM2015085 | Alžběta Türková<br>Ivo Kabelka<br>Tereza Králová<br>Lukáš Sukeník<br>Robert Vácha |
| Ministry of Education, Youth and Sports of the Czech Republic | LM2015042 | Alžběta Türková<br>Ivo Kabelka<br>Tereza Králová<br>Lukáš Sukeník<br>Robert Vácha |

The funders had no role in study design, data collection and interpretation, or the decision to submit the work for publication.

## Author contributions

Alzbeta Tuerkova, Conceptualization, Data curation, Formal analysis, Investigation, Visualization, Methodology; Ivo Kabelka, Conceptualization, Data curation, Formal analysis, Validation, Investigation, Methodology; Tereza Králová, Resources, Data curation, Methodology; Lukáš Sukeník, Data curation, Software, Validation, Methodology; Šárka Pokorná, Data curation, Validation, Methodology; Martin Hof, Conceptualization, Resources, Funding acquisition, Methodology; Robert Vácha, Conceptualization, Resources, Software, Supervision, Funding acquisition, Project administration

## Author ORCIDs

Alzbeta Tuerkova (iD) https://orcid.org/0000-0001-7235-9029
Martin Hof (iD) https://orcid.org/0000-0003-2884-3037
Robert Vácha (iD) https://orcid.org/0000-0001-7610-658X

## Decision letter and Author response

Decision letter https://doi.org/10.7554/eLife.47946.sa1
Author response https://doi.org/10.7554/eLife.47946.sa2

## Additional files

### Supplementary files

• Transparent reporting form

### Data availability

All data generated or analysed during this study are included in the manuscript and supporting files. Source data files have been provided for Figures 1 -11.

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

## Appendix 1

**Appendix 1—table 1. Combination of parameters for O-PSC-NE model peptides with *rigid* and *flexible* kink.** Effect of peptide length (4.0, 5.0, or 7.0 nm), width of hydrophobic sector (150˚, 190˚, 180˚, 230˚, 270˚, 310˚, 350˚), on the formation of different pore structures is depicted.

| | Hydrophobic sector [˚] | | | | | | | | | | | |
| | 150 | | 190 | | 230 | | 270 | | 310 | | 350 | |
| Length [nm] | flexible | rigid | flexible | rigid | flexible | rigid | flexible | rigid | flexible | rigid | flexible | rigid |
|---|---|---|---|---|---|---|---|---|---|---|---|---|
| 4.0 | – | – | – | – | – | – | – | – | – | BP | I/A | I/A |
| 5.0 | – | – | – | – | – | – | TP | BP | I/A | BP | I/A | I/A |
| 7.0 | TP | S/A | TP | TP | TP | DTP | I/A | DTP | I/A | I/A | I/A | I/A |

Toroidal pore (TP); Disordered toroidal pore (DTP); Barrel-stave pore (BP); Hourglass pore (HG); U-shaped pore (U); peptide insertion into the membrane without aggregation (I/A); peptide aggregation in solvent (S/A); no membrane binding (–)

**Appendix 1—table 2.** Number of distinct starting configurations for MARTINI simulations.

| | | | Simulations | | |
| Peptide | Variant | Membrane | Count† | Length [μs] | Figure‡ |
|---|---|---|---|---|---|
| LL-37 | WT | POPC | 2 | 45, 45 | *Figure 7* |
| | WT* | POPC | 1 | 67 | – |
| | G14L | POPC | 1 | 90 | *Figure 7* |
| | I13G/G14L | POPC | 1 | 67 | *Figure 7* |
| Buforin II | WT | POPC | 3 | 45, 41, 14 | – |
| | WT* | POPC | 3 | 45, 33, 21 | – |
| | P11L | POPC | 2 | 34, 27 | *Figure 4*, *Figure 5*, *Figure 6* |
| | P11L | POPC:POPG (1:1) | 1 | 45 | – |
| | P11A | POPC:POPG (1:1) | 1 | 45 | – |
| | P11G | POPC:POPG (1:1) | 1 | 45 | – |
| | P11G* | POPC:POPG (1:1) | 1 | 45 | – |
| | WT | POPC:POPG (1:1) | 1 | 45 | – |
| | WT* | POPC:POPG (1:1) | 1 | 45 | – |
| | P11L | POPC:POPS (1:1) | 1 | 45 | *Figure 5*, *Figure 6* |
| | P11A | POPC:POPS (1:1) | 1 | 45 | – |
| | P11G | POPC:POPS (1:1) | 1 | 45 | – |
| | P11G* | POPC:POPS (1:1) | 1 | 45 | – |
| | WT | POPC:POPS (1:1) | 1 | 45 | – |
| | WT* | POPC:POPS (1:1) | 1 | 45 | – |
| | P11A | POPC | 2 | 42, 27 | – |
| | P11G | POPC | 1 | 12 | *Figure 4* |
| | P11G* | POPC | 2 | 45, 39 | – |
| | P11G⁻ | POPC | 2 | 45, 17 | – |
| Magainin 2 | WT | POPC | 2 | 45, 45 | – |

*Appendix 1—table 2 continued on next page*

*Appendix 1—table 2 continued*

|  |  |  | Simulations |  |  |
|---|---|---|---|---|---|
|  | WT* | POPC | 4 | 45, 45, 45, 45 | *Figure 9* |
|  | G13P | POPC | 2 | 45, 45 | – |
|  | G13P* | POPC | 2 | 45, 45 | – |
| δ-lysin | WT | POPC | 1 | 44 | *Figure 9* |
|  | WT* | POPC | 1 | 44 | – |
| CandK | WT | POPC | 2 | 45, 35 | – |
|  | WT* | POPC | 1 | 49 | – |
| CandKR | WT | POPC | 4 | 45, 12, 10, 10 | *Figure 8* |
|  | P14Q/Q15P | POPC | 1 | 38 | *Figure 8* |

† Number of distinct starting configurations.

‡ List of figures in the main text where representative snapshots of these systems appear.

\* a sequence where glycine/proline (usually kink-forming residues) was forced into an $\alpha$-helix.

ˆ a sequence where only single residue formed the flexible kink.

**Appendix 1—table 3.** Average number of solvent molecules in the region from $-0.9$ to $0.9$ nm from membrane center in the last 10 μs.

| Peptide | Variant | Membrane | Mean | Standard deviation |
|---|---|---|---|---|
| – | – | POPC | 0.7 | 1.1 |
| LL-37 | WT | POPC | 27.4 | 11.9 |
|  | WT* | POPC | 9.8 | 6.1 |
|  | G14L | POPC | 15.3 | 7.8 |
|  | I13G/G14I | POPC | 24.6 | 8.6 |
| Buforin II | WT | POPC | 10.5 | 7.6 |
|  | WT* | POPC | 27.6 | 10.3 |
|  | P11L | POPC | 43.7 | 17.8 |
|  | P11A | POPC | 23.1 | 8.6 |
|  | P11G | POPC | 7.7 | 6.0 |
|  | P11G* | POPC | 21.4 | 7.4 |
|  | P11Gˆ | POPC | 20.0 | 14.8 |
|  | WT | POPC:POPG (1:1) | 4.0 | 3.2 |
|  | WT* | POPC:POPG (1:1) | 6.0 | 3.6 |
|  | P11L | POPC:POPG (1:1) | 14.9 | 8.0 |
|  | P11A | POPC:POPG (1:1) | 5.3 | 3.6 |
|  | P11G | POPC:POPG (1:1) | 3.7 | 3.2 |
|  | WT | POPC:POPS (1:1) | 3.8 | 3.0 |
|  | WT* | POPC:POPS (1:1) | 7.8 | 4.0 |
|  | P11L | POPC:POPS (1:1) | 5.4 | 3.6 |
|  | P11A | POPC:POPS (1:1) | 7.7 | 4.9 |
|  | P11G | POPC:POPS (1:1) | 3.9 | 3.3 |
| Magainin 2 | WT | POPC | 16.1 | 6.8 |
|  | WT* | POPC | 30.1 | 8.9 |
|  | G13P | POPC | 60.3 | 18.8 |

*Appendix 1—table 3 continued on next page*

*Appendix 1—table 3 continued*

| Peptide | Variant | Membrane | Mean | Standard deviation |
|---------|---------|----------|------|--------------------|
|         | G13P*   | POPC     | 45.3 | 11.5               |
| δ-lysin | WT      | POPC     | 15.8 | 6.8                |
|         | WT*     | POPC     | 12.4 | 5.7                |
| CandK   | WT      | POPC     | 6.9  | 3.4                |
|         | WT*     | POPC     | 6.0  | 4.3                |
| CandKR  | WT      | POPC     | 7.1  | 4.2                |
|         | P14Q/Q15P | POPC   | 6.6  | 4.5                |

\* a sequence where glycine/proline (usually kink-forming residues) was forced into an α-helix.
ˆa sequence where only single residue formed the flexible kink.
Lipid ratio is expressed as mol/mol.

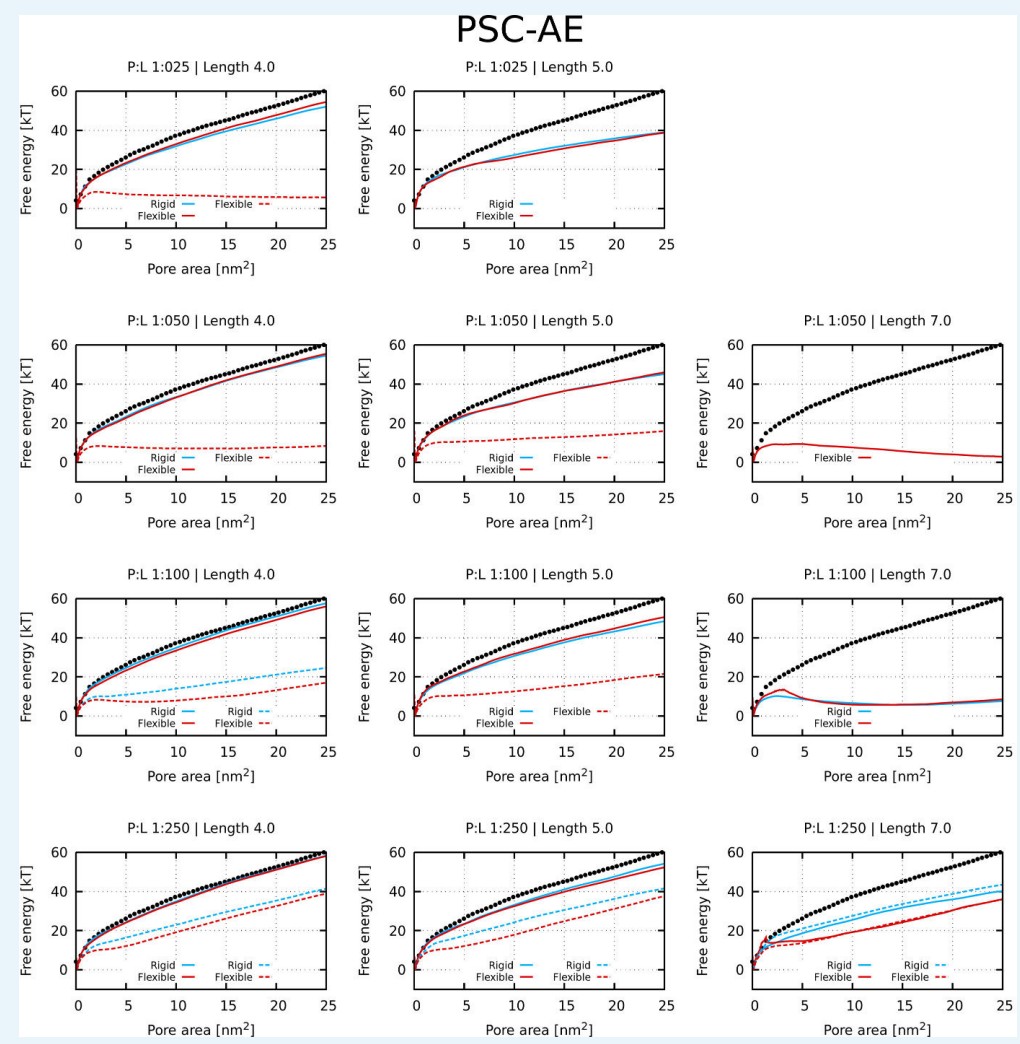

**Appendix 1—figure 1.** Free energy profiles of pore formation by PSC-AE peptides. Standard and more hydrophobic PSC-AE peptides are shown in full and dashed lines, respectively. The

dotted line corresponds to the free energy profile of our reference system, a pure membrane without any peptides.

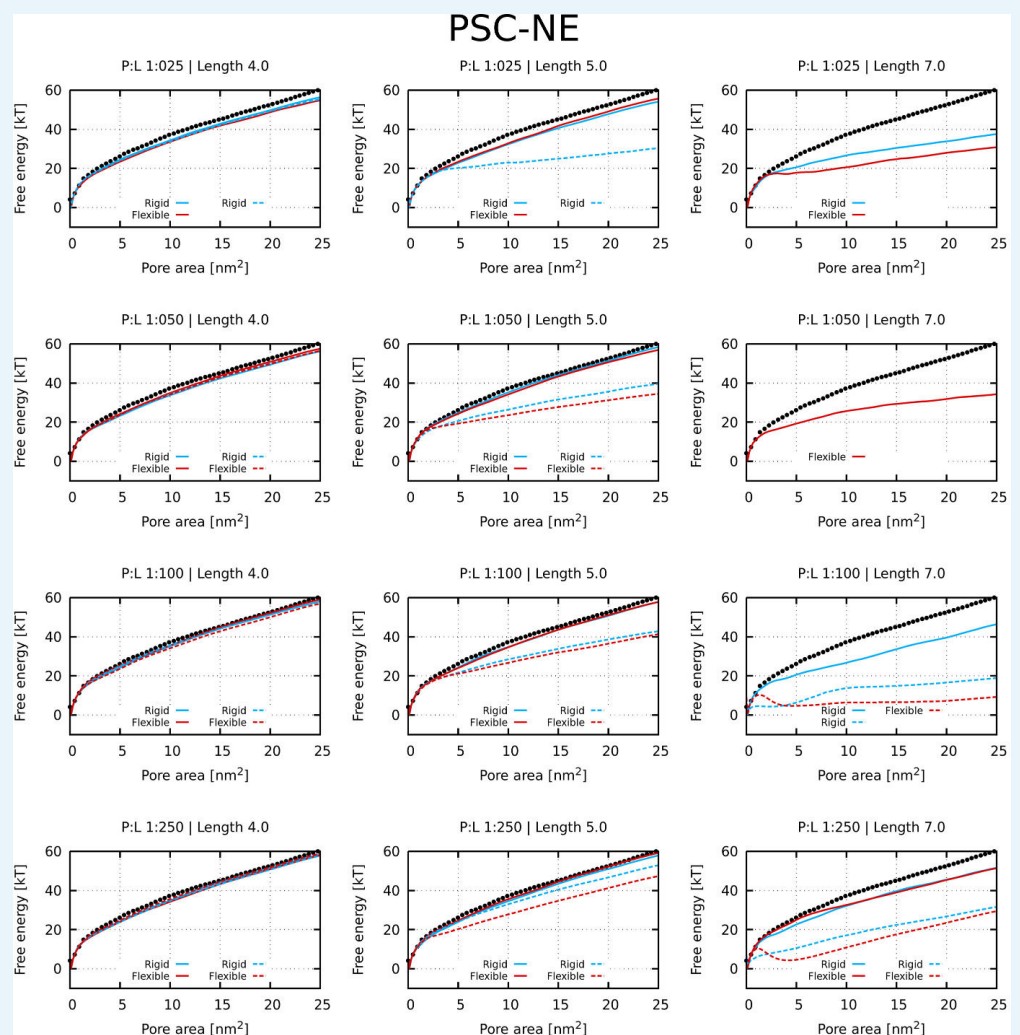

**Appendix 1—figure 2.** Free energy profiles of pore formation by PSC-NE peptides. Standard and more hydrophobic PSC-NE peptides are shown in full and dashed lines, respectively. The dotted line corresponds to the free energy profile of our reference system, a pure membrane without any peptides.

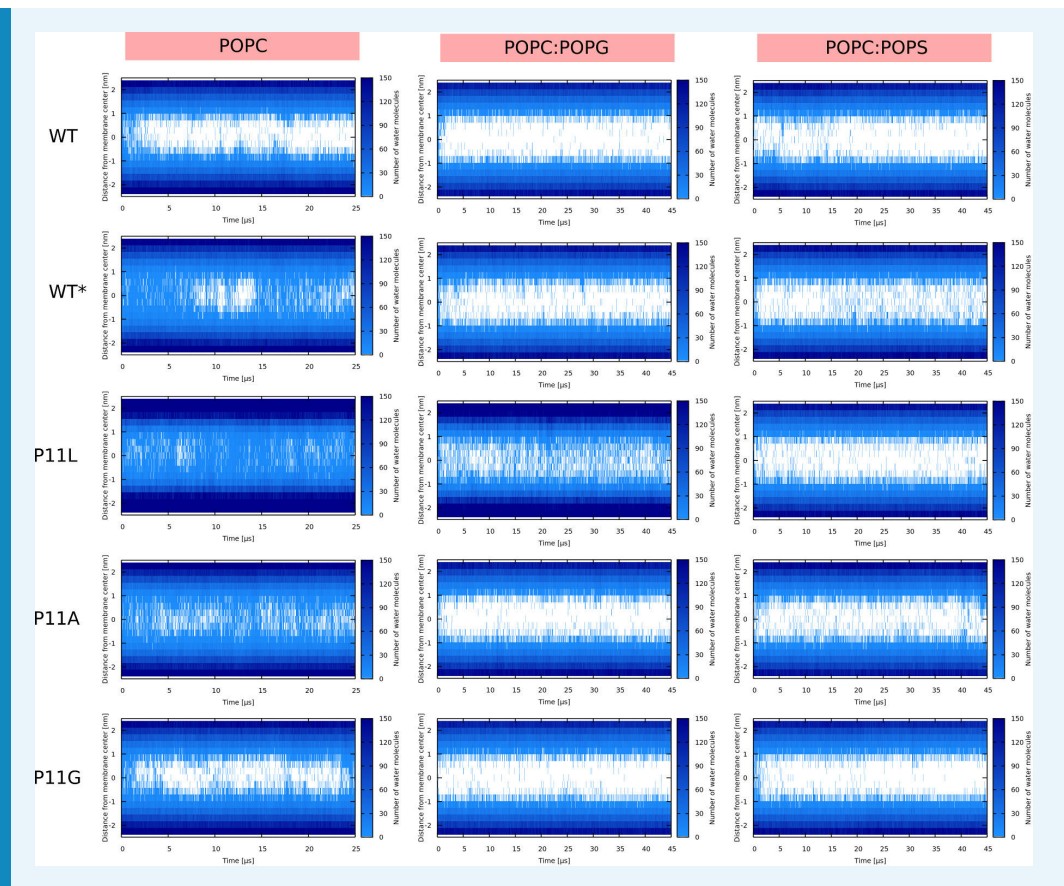

**Appendix 1—figure 3.** Water density profiles for Buforin II peptides. Note that WT* stands for wild-type sequence where proline residue was forced to be α-helical.

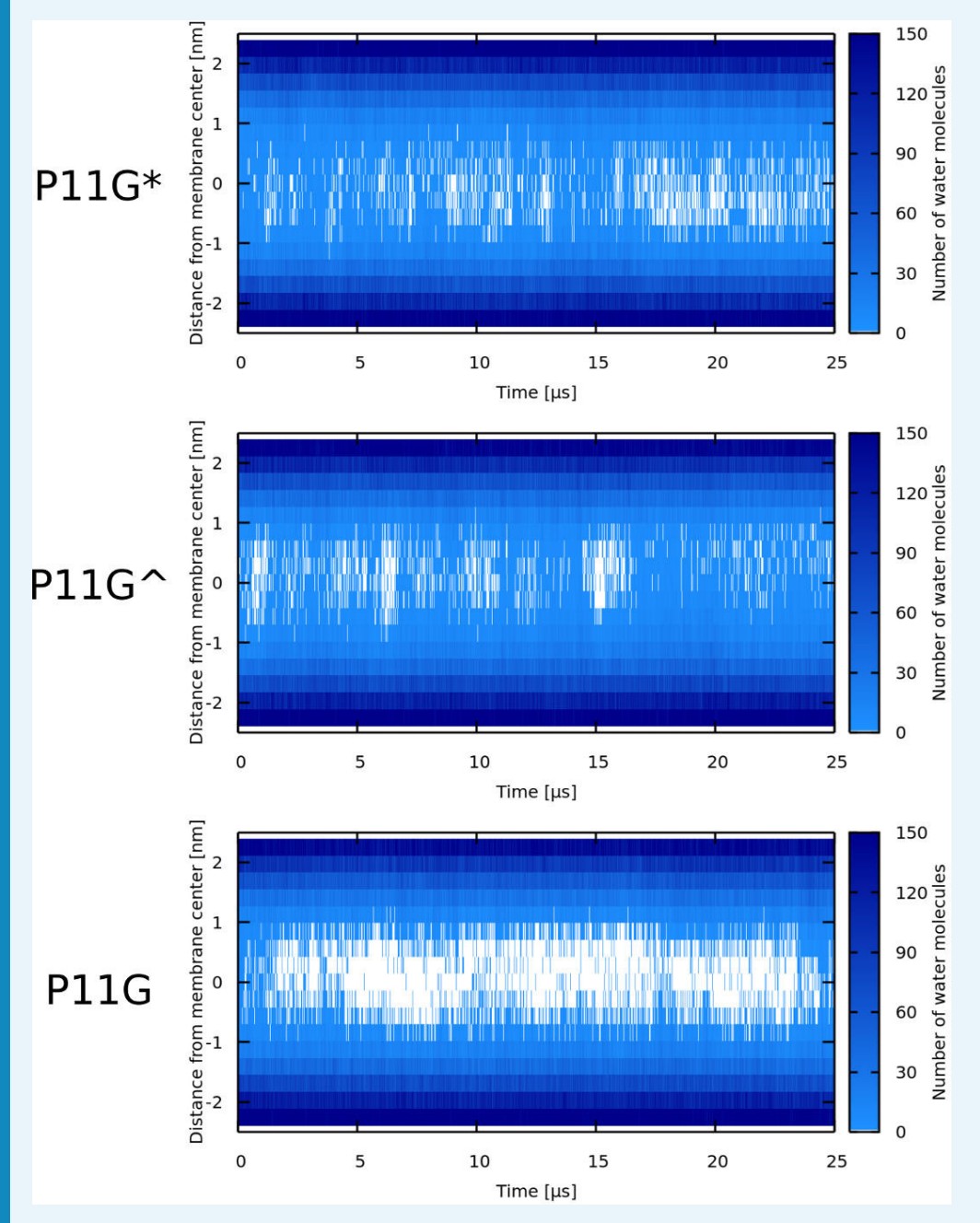

**Appendix 1—figure 4.** Water density profiles for Buforin II P11G in POPC membrane. P11G* denotes a sequence that was forced to be α-helical. In case of Buforin II P11G͠, only the glycine residue formed the flexible kink.

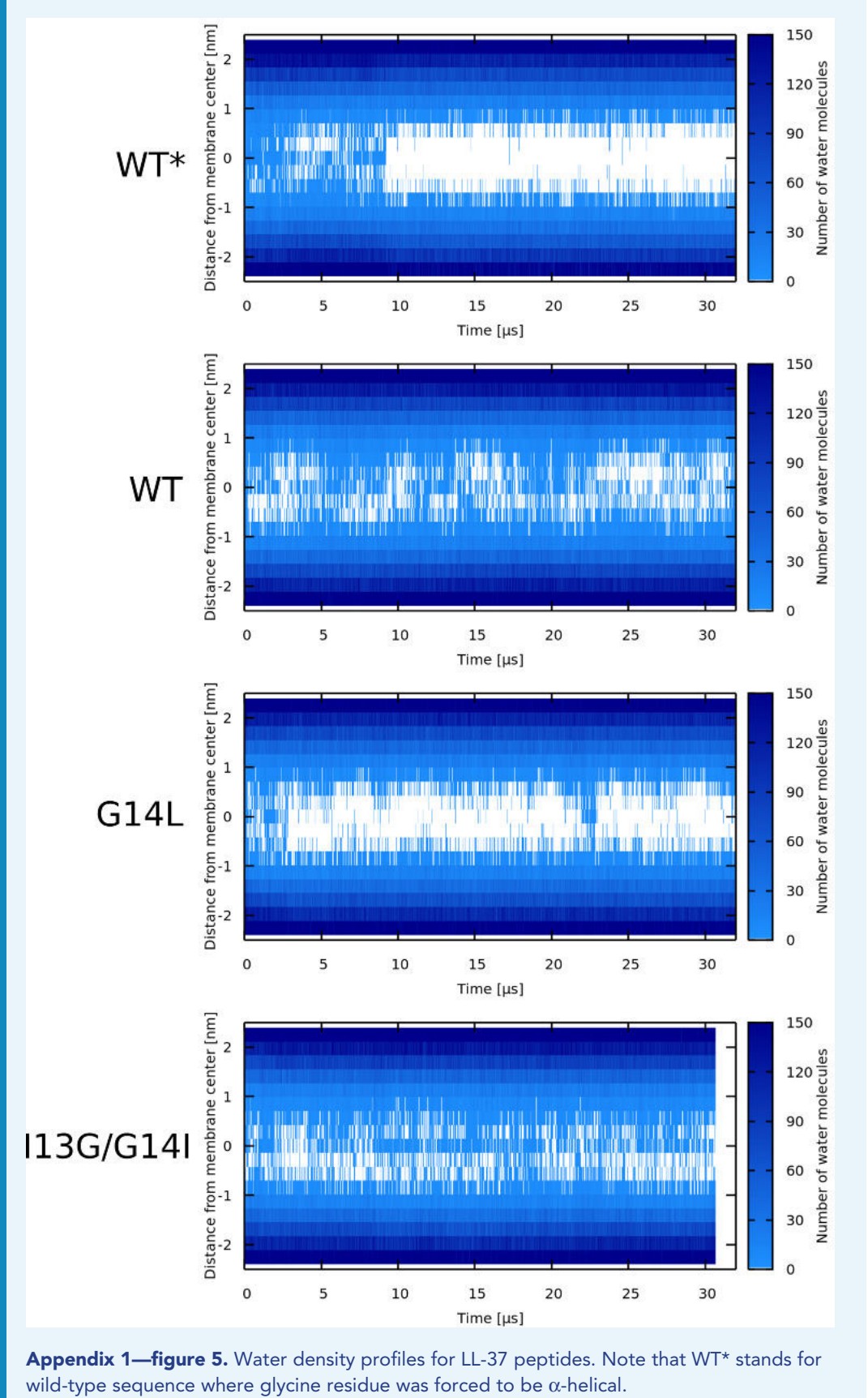

**Appendix 1—figure 5.** Water density profiles for LL-37 peptides. Note that WT* stands for wild-type sequence where glycine residue was forced to be α-helical.

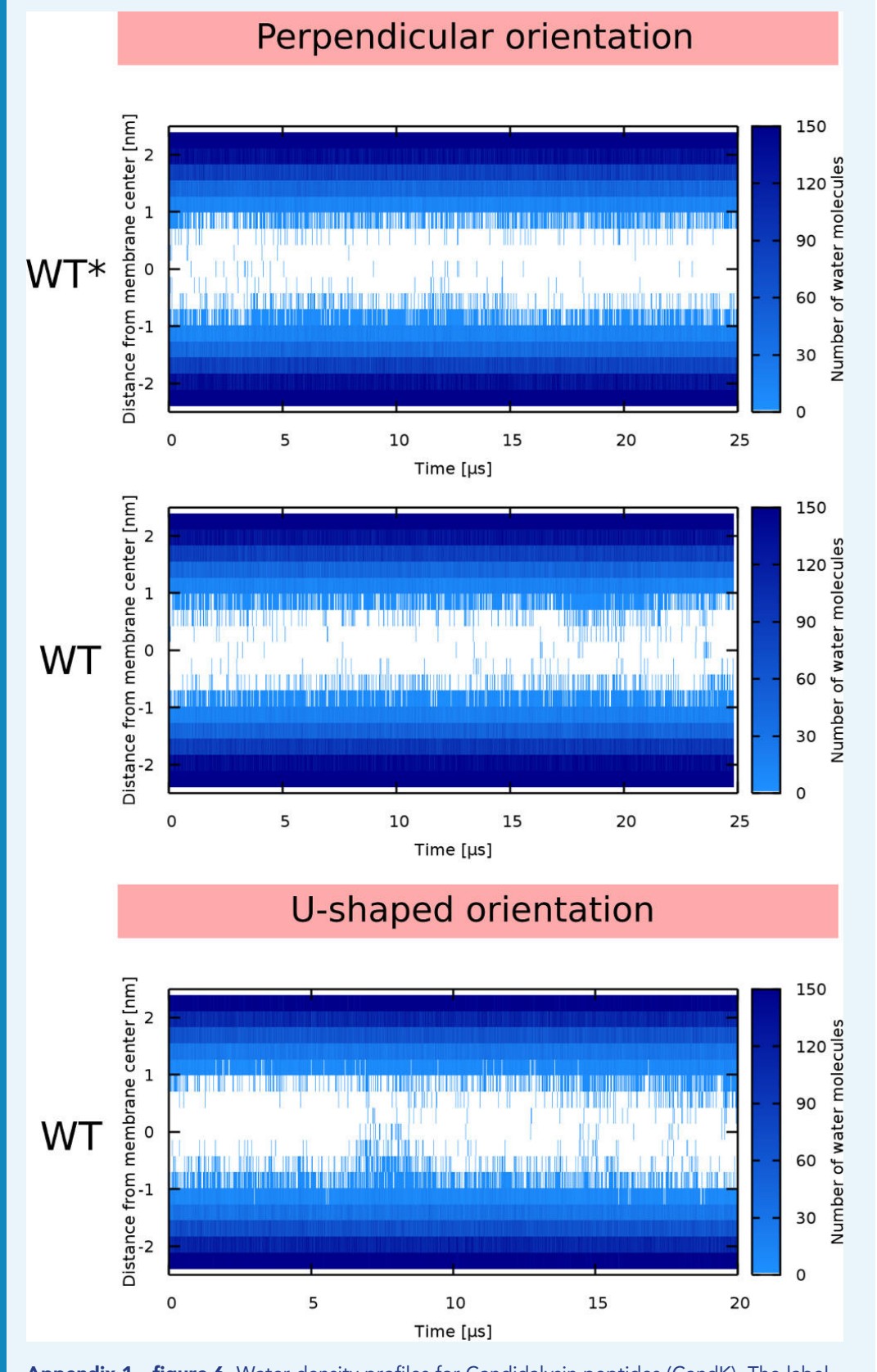

**Appendix 1—figure 6.** Water density profiles for Candidalysin peptides (CandK). The label describes the starting configuration of the simulation. Note that WT* stands for wild-type sequence where proline residue was forced to be α-helical.

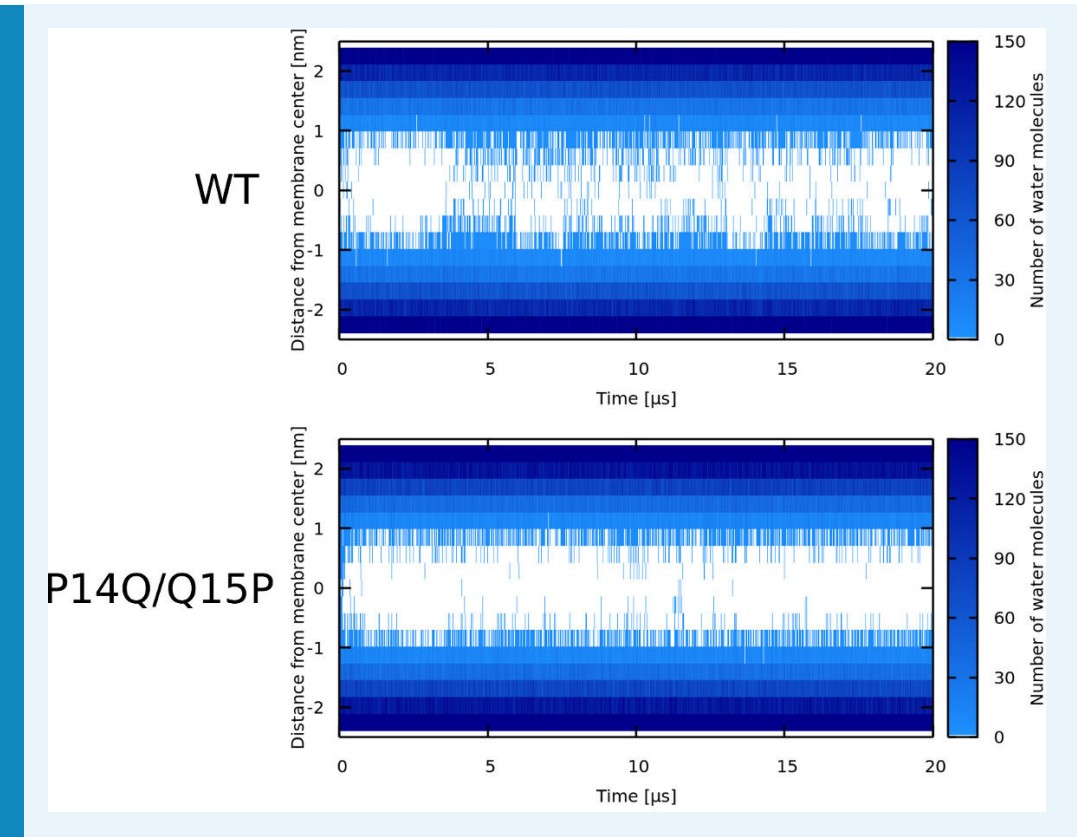

**Appendix 1—figure 7.** Water density profiles for Candidalysin peptides (CandKR). Note that WT* stands for wild-type sequence where proline residue was forced to be α-helical.

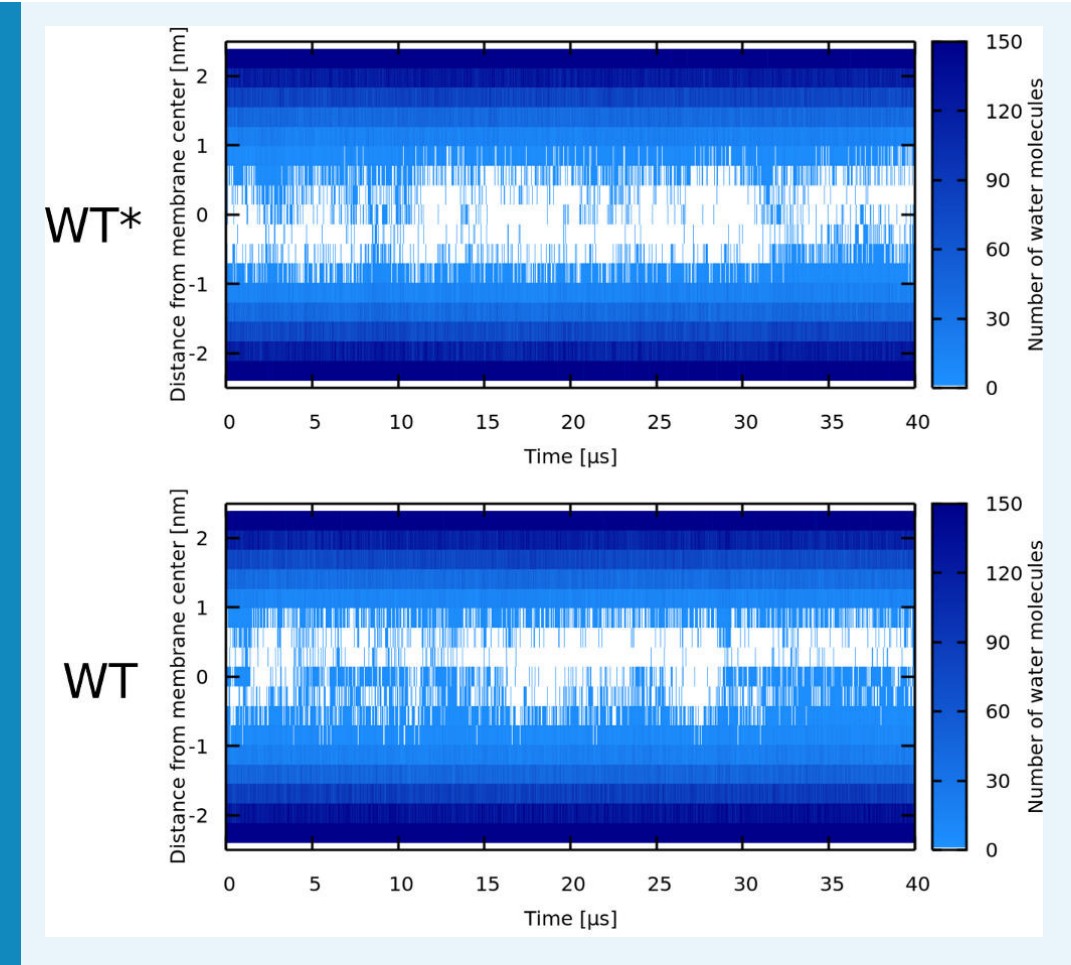

**Appendix 1—figure 8.** Water density profiles for δ-lysin peptides. Note that WT* stands for wild-type sequence where glycine residue was forced to be α-helical.

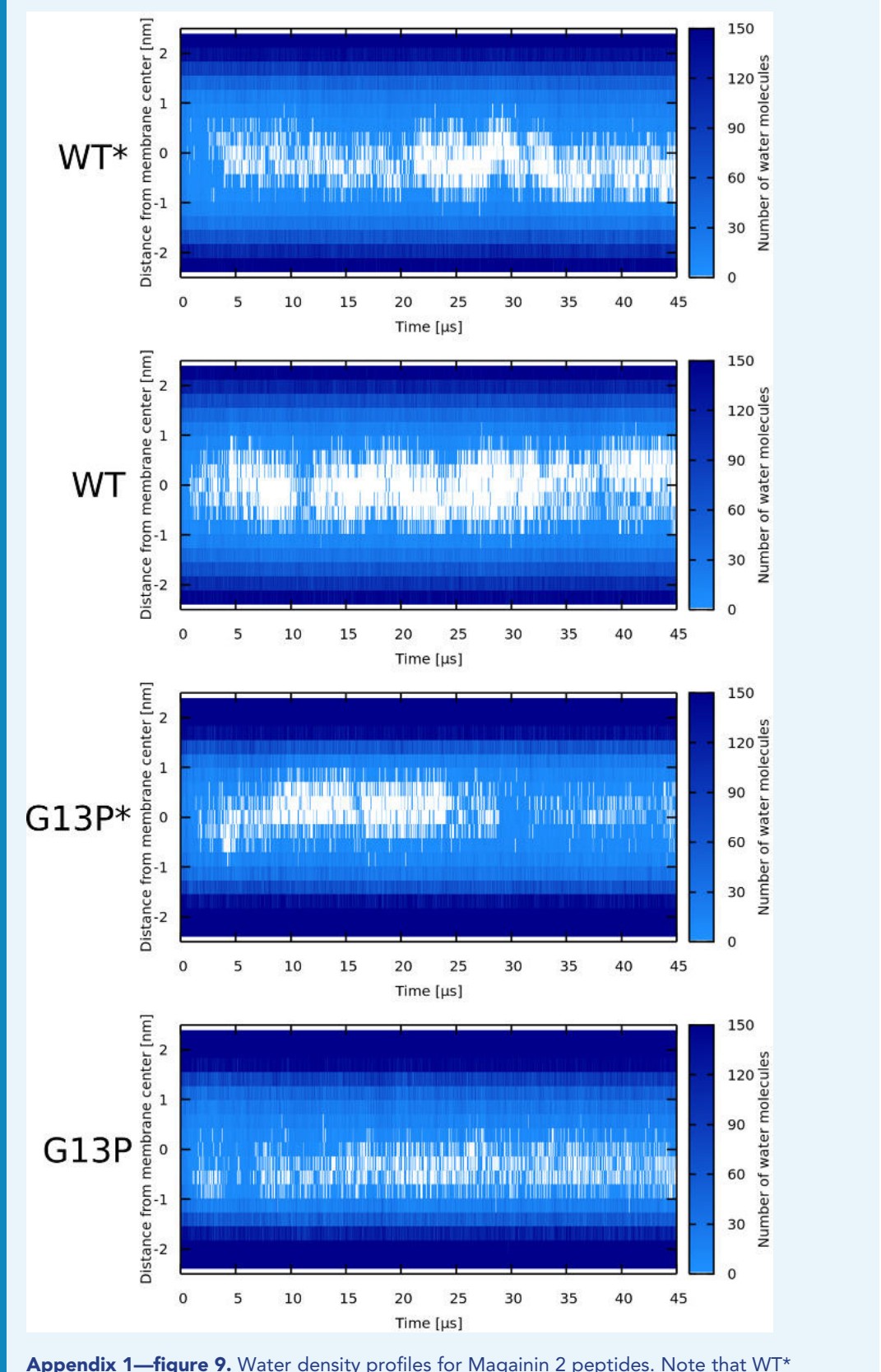

**Appendix 1—figure 9.** Water density profiles for Magainin 2 peptides. Note that WT* and G13P* stand for wild-type sequences where kink-forming residues were forced to be α-helical.

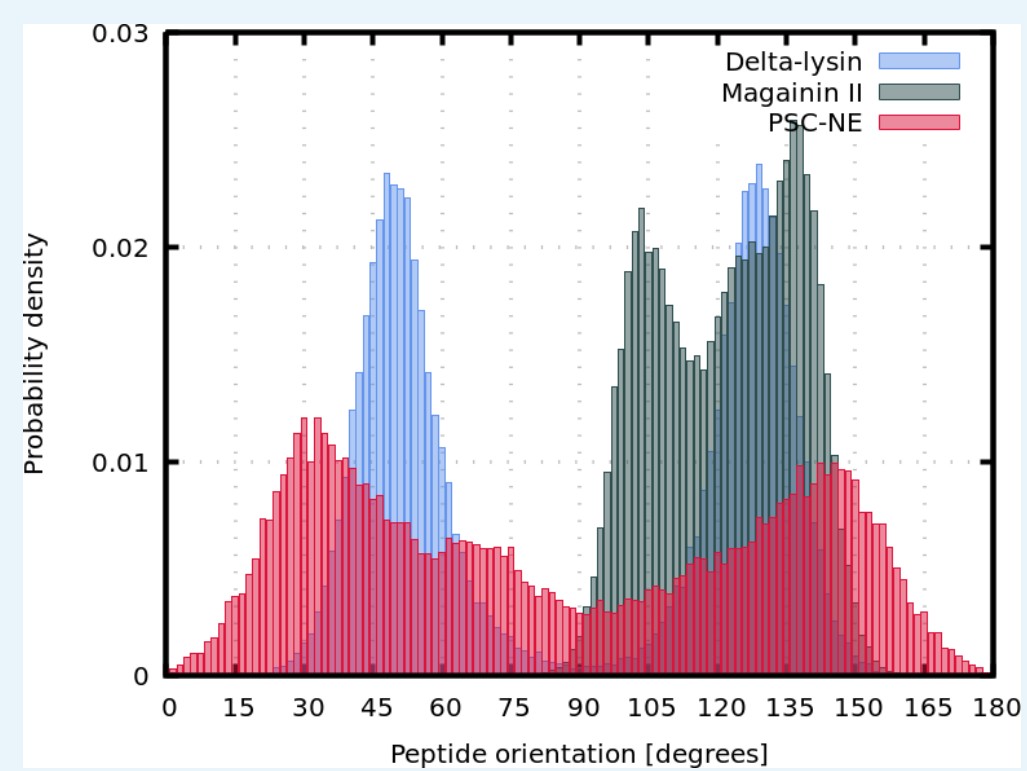

**Appendix 1—figure 10.** Comparison of angles between peptides and z-axis in systems with synergistic interactions. Phenomenological (PSC-NE) models (red), Magainin 2 peptides (green), and δ-lysin peptides (blue).

