## [Decision Letter]

**Acceptance summary:**

Designing more efficient antimicrobial peptides to take on antibiotic-resistant bacteria is an important challenge for health care. This paper presents results from simulations and in vitro experiments showing the effect of a kink in the structure of the peptide on the architecture of the pores they form in membranes and their ability to induce membrane leakage. This work may inspire the design of new drugs to replace antibiotics.

**Decision letter after peer review:**

[Editors’ note: the authors submitted for reconsideration following the decision after peer review. What follows is the decision letter after the first round of review.]

Thank you for submitting your work entitled "Effect of helical kink in antimicrobial peptides on membrane pore formation" for consideration by *eLife*. Your article has been reviewed by three peer reviewers, and the evaluation has been overseen by a Reviewing Editor and a Senior Editor. The reviewers have opted to remain anonymous.

We think that the combination of 2 types of simulation and experiments is a positive aspect of your work. But we also have major concerns that prevent considering your paper for publication in *eLife* since we believe that the revisions would take more than2 months. As explained in the detailed reports, your manuscript must better explain your approach and the procedures used in simulations. The simulations have to be repeated many times to be convincing and we had the impression that they were performed only once. Moreover, you should also perform simulations where flexibility of the AMPs is allowed, since modeling AMPs with a fixed kink appears unrealistic. We also consider that more modern model for the peptides (peptides parallel to the surface, pores and lysis due to the disordering of lipids) should be considered or at least, discussed.

Would you decide to resubmit your manuscript to *eLife* in the future, we recommend to seriously take into account all the suggestions and questions from the reviews.

Reviewer #1:

Overall the work is logically carried out and clearly described but we feel that there are some major concerns and the experimental validation is relatively limited.

It does not seem clear that the detail of the first very coarse-grained MD is sufficient to capture the behaviour of the system – no validation of the model is presented. In addition, the description of the model simply mentions 'length, hydrophobicity and flexibility of the link'. How is the hydrophobicity encoded? The Results and Discussion confusingly talks about a patch of 180degrees.

Neither kind of MD seems to have been subjected to repetitions (necessary for a Monte Carlo process) to see if the results are repeatable between different simulations

The membrane composition that was used is POPC:POPG in 1:1 ratio. This is highly charged and not in agreement with experiments and other studies mainly using POPC:POPG or POPE:POPG in 4:1 or 3:1 ratio as bacterial model membrane. The authors should provide a reference for this 1:1 POPC:POPG composition.

A further problem is related to the AMP models used in the study. While the authors specify that the flexibility of kink in the super CG model was systematically varied, they do not provide details such as angles/rotations between the two helices that were tested. In Martini models, secondary structure is fixed, and 4 heavy atoms forming one bead is a standard. However, the kink was reported to be modelled as a random coil. The question is how many residues are defined to form this kink, and again, it is not clear here what angles/rotations were used between the two helices. These models are then incorporated into the membrane, but since the AMPs are Martini models, this doesn't allow the relaxation of the structures, meaning the authors can't sample enough conformations because of the rigid backbone. The unflexible AMPs leading to barrel-stave is most likely an artifact of the model. A study by Yang et al., 2001, concluded that "among naturally produced peptides that we have investigated, only alamethicin conforms to the barrel-stave model. Other peptides, including magainins, melittin and protegrins, all appear to induce transmembrane pores that conform to the toroidal model". While both Melittin and Magainin do have a Gly residue in the middle part of the sequence, the structure of Magainin in DPC micelles doesn't show a kink. Nevertheless, the authors claim that it is the flexible peptide kink in the helical structure that causes the instability of barrel pores (BPs). To conclude, the authors should have parametrised AMP models against an atomistic model in order to get the correct dynamics.

The experimental results in Figure 5 are valuable are limited to mutations at one residue (A) and seem to show no significant difference (this should be tested statistically) in (B)

There should be some justification for the very large pores in Figure 6.

Reviewer #2:

The manuscript presents results from simulations and experiments performed to study effects that helical kinks in antimicrobial peptides have on the ability of these peptides to produce pores in phospholipid membranes. Most of the manuscript describes many different simulations using two levels of coarse-grained models (very coarse phenomenological and some more detailed, using MARTINI force field) that were performed in order to reach the conclusion that kinks disrupts the stability of barrel-stave pores, while stabilizing toroidal pores. Combining simulations with experiment strengthens the arguments obtained from the simulations.

The weakness of the manuscript is in the description of the simulation work done. Thus, for example, to help the reader to understand the phenomenological model and the results from simulations with it, the authors should provide a brief description of this model, since, as I suspect the model used may be not as familiar to the reader as MARTINI force field. This description can be given in Supplementary Material. In some places it was hard for me to follow the paper because of its non-sequential order of referring to Figures and Tables (for example, see the Results were the reader is sent to figures 5D and S13 before discussion of previous figures). I also did not understand how many different initial configurations and arrangements were considered for each kind of peptide in MARTINI simulations. This is a very important issue, since simulations were done starting with preformed pores. The authors mention that the simulations were performed starting from different arrangements; see subsection “MARTINI simulations” and they refer to figure S1, that depicts a snapshot from simulation with phenomenological model. Also, it would be nice to know the length of each MARTINI simulation and the final outcome (pore morphology). Perhaps the authors can extend Table 1 to add useful information. Some of the Figures in Supplementary Material may be redundant. Thus, there is no reference in the text to Figures S3 and S4.

Summary: Interesting work, but the manuscript requires serious editing to streamline the arguments and make the paper more readable.

Reviewer #3:

The paper by Türkova et al. describes some low-resolution MD simulations of the effects of the kinks in AMPs on their pore-forming activity. The simulations give some interesting insights in particular as many interesting information can be obtained by very simple phenomenological models that merely represent hydrophobic / hydrophilic distributions. On the other hand, even the Martini model is probably not suitable to reproduce some experimental details (cf. below).

Therefore, the authors need to be clearer about such details and discuss in depth a number of issues. Otherwise, the wrong conclusions can easily be drawn from the nice almost atomistic views shown in the paper.

For example, it is of importance to underline that pore structures have been used as the starting structures for the simulations and that the analysis only compares which of the supposed models is more stable. A more convincing approach would have been to take a random distribution of peptides and check out if they assemble into stable or even transient pore structures, but this may not be possible even with the approaches choses that permit extended simulation times. As a matter of fact, all energies found for the various pore structures in the paper have positive energies and are unfavorable (when compared to the absence of a pore). As a consequence, none of the assumed models should represent the pore structures observed by experimental methods. This should be clearly mentioned and underlined. The paper would much increase in merit from such an open and unbiased disussion.

Other models such as the carpet or the SMART model have not been discussed in the paper. There is good evidence from a number of experimental observations that neither the barrel stave nor the toroidal pore model represent well the activities of these peptides. The results of the paper seem to support such alternative views.

As for the synergistic activities, similar considerations apply. An interesting observation is that in POPC or POPC/POPG membranes magainin and PGLa are both oriented parallel to the membrane surface in a stable manner. Only in membranes made of saturated lipids was membrane insertion of PGLa observed. It should be discussed to which extent the Martini model used here can represent these special properties of the lipid fatty acyl chains. Of course, it would be highly rewarding to model this different behavior of PGLa in the presence of magainin.

In my opinion these are interesting data which should be presented in a different manner to further increase the merit of the paper.

[Editors’ note: further revisions were suggested prior to acceptance, as described below.]

Thank you for resubmitting your work entitled "Effect of helical kink in antimicrobial peptides on membrane pore formation" for further consideration by *eLife*. Your revised article has been evaluated by Olga Boudker (Senior Editor) and a Reviewing Editor.

The manuscript has been improved but there are some remaining issues that need to be addressed before acceptance, as outlined below:

– Justify in the text the choice of membrane model, and its relevance towards living cells. Comment on the potential consequences of the high charge of the membrane.

– Clearly justify the large pores in the text.

– Add some sentences in the text to roughly define the methods, in addition to the Methods and Materials section.

– Check the organization of the text to make it more linear.

– Use a more obvious notation for the figure names.

Reviewer #2:

This paper presents a description of computer simulations that study a class of antimicrobial peptides (AMP) containing a sharp bend (kink) in their regular helical structure. The simulations are performed using two types of coarse-grained description of the systems: a very coarse-grained phenomenological model and a more detailed coarse-grained model-MARTINI. The goal is to understand the effect the kink has on the ability to form leaky pores in model membranes. The main question asked is how much of general information can one obtain from coarse-grained models and can this information help us to understand general issues related to AMP action such as morphology and stability of the pores. In my opinion, the authors accomplished their goal. Both coarse-grained models and supporting their conclusion experiments on leakage provided "molecular rationalization of the effect of a proline /glycine kink in helical peptides in the context of membrane pore formation."

The authors did a pretty good job of rewriting the manuscript. There are still small issues that make reading somewhat difficult in certain places: the reader needs to move forward from the main text to the section on Methods and Materials and back to understand clearly what is going on. A few sentences placed in the main text about methods can make the flow of reading much smoother. The authors should carefully read the manuscript and see how they can make the reading of the text smoother, so there is no need to go back and forth while reading the paper. Some very small remarks can be helpful; e.g., I got initially confused by notation used to denote figures such as Figure 1A and Figure A1. Just writing Figure A1 (see Appendix) can be very helpful.

Some other issues: 1) I did not understand what the free energy curve from Figure 1 denoted by dots stands for. 2) In subsection “Effect of Peptide Flexibility and Length on the Structure of Membrane Pores” the authors say that bias was applied to lipids to observe pore formation. What kind of bias?

Conclusion: The paper accomplished the goal of explaining the role of kinks as seen from the coarse-grained model and confirming leakage experiments. I recommend publication after some editing in response to remarks from above.

Reviewer #4:

I think most of the reviewer's comments have been adequately addressed. However, I do feel strongly that the authors should comment on their choice of membrane model and how it relates to in vivo compositions and any potential consequences of using such a highly charged membrane. Furthermore, the reviewers point about the justification of very large pores is not addressed to an appropriate level of detail. Justification for these pores should be clearly included in the manuscript.

---

## [Author Response]

Reviewer #1:Overall the work is logically carried out and clearly described but we feel that there are some major concerns and the experimental validation is relatively limited.It does not seem clear that the detail of the first very coarse-grained MD is sufficient to capture the behaviour of the system – no validation of the model is presented. In addition, the description of the model simply mentions 'length, hydrophobicity and flexibility of the link'. How is the hydrophobicity encoded? The Results and Discussion confusingly talks about a patch of 180degrees.

We rewrote the whole manuscript including the description and previous use of the employed models. The first coarse-grained model (phenomenological model) has been developed in 2011 (Vácha and Frenkel,2011) to capture the behavior of amphiphilic helical peptides and has been validated to the existing structures. Since then, we have shown that the model provides valuable data for peptide translocation and pore formation, which has already been experimentally verified (Vácha and Frenkel,2013; Leber et al.,2018). The details of the model are described in brief in the Materials and methods sections now. We agree that the description of the hydrophobic content could have been misleading for a reader. Commonly, the distribution of residues in an α-helix is visualized using the helical wheel projection. We have adopted this description and now we express the hydrophobic content as the angle of a hydrophobic sector. For example, the peptide with the hydrophobic sector of 180 degrees is half hydrophilic and half hydrophobic.

Neither kind of MD seems to have been subjected to repetitions (necessary for a Monte Carlo process) to see if the results are repeatable between different simulations

The information about repetitions of the simulations was not obvious or mistakenly omitted from the previous manuscript. With the MARTINI model, we have first tested the preferred orientation of the peptides in a pre-formed pore of a fixed size starting from various initial conditions. The number of different peptide orientations/conformations at the end of these simulations determined the number of subsequent simulations starting from those distinct configurations. Despite the different starting configurations, the observed behavior was consistent for all but one peptide variant as described in the manuscript.

In the Monte Carlo simulations, an enhanced sampling method (Wang-Landau) was employed to sample dozens of pore opening and closing events. These simulations are thus very time consuming due to the thorough sampling of all the accessible configurations. In other words, instead of employing number of simulations each capturing one pore opening and closing, this method requires dozens of such events within one simulation for convergence.

This information is now clearly stated in the Materials and methods section of the manuscript.

The membrane composition that was used is POPC:POPG in 1:1 ratio. This is highly charged and not in agreement with experiments and other studies mainly using POPC:POPG or POPE:POPG in 4:1 or 3:1 ratio as bacterial model membrane. The authors should provide a reference for this 1:1 POPC:POPG composition.

The POPC:POPG 1:1 lipid ratio was used in one of the first studies of the Buforin II peptide (DOI: 10.1021/bi0004549). We included this information in the manuscript. Furthermore, we would like to kindly point out that this proportion of charged lipids is not excessive in general. For some bacteria (e.g., *B. subtilis*), the amount of negatively charged lipids was reported to exceed 70% with majority of phosphoglycerol (PG) (DOI: 10.1128/jb.168.1.334-340.1986).

A further problem is related to the AMP models used in the study. While the authors specify that the flexibility of kink in the super CG model was systematically varied, they do not provide details such as angles/rotations between the two helices that were tested. In Martini models, secondary structure is fixed, and 4 heavy atoms forming one bead is a standard. However, the kink was reported to be modelled as a random coil. The question is how many residues are defined to form this kink, and again, it is not clear here what angles/rotations were used between the two helices. These models are then incorporated into the membrane, but since the AMPs are Martini models, this doesn't allow the relaxation of the structures, meaning the authors can't sample enough conformations because of the rigid backbone. The unflexible AMPs leading to barrel-stave is most likely an artifact of the model. A study by Yang et al. (Biophys J. 2001 Sep; 81(3): 1475-1485) concluded that "among naturally produced peptides that we have investigated, only alamethicin conforms to the barrel-stave model. Other peptides, including magainins, melittin and protegrins, all appear to induce transmembrane pores that conform to the toroidal model". While both Melittin and Magainin do have a Gly residue in the middle part of the sequence, the structure of Magainin in DPC micelles doesn't show a kink. Nevertheless, the authors claim that it is the flexible peptide kink in the helical structure that causes the instability of barrel pores (BPs). To conclude, the authors should have parametrised AMP models against an atomistic model in order to get the correct dynamics.

We thank the reviewer for raising this important point.

In the phenomenological model, we have considered the kink to be either rigid (i.e., fixed with no way to bend) or fully flexible (free to adopt any orientation). For clarity, the information about the range of motion of the helical segments was added in Figure 10.

To clarify the effect of the peptide kink, we studied peptides with proline/glycine residues in their sequences.

In the MARTINI simulations, we considered the central proline/glycine residue and one neighbouring residue on either side to be flexible. In case of Buforin II P11G, we have additionally considered the case of only one kink-forming residue. To further investigate the effect of peptide flexibility, we simulated the same sequences and forced the peptides into a fully helical secondary structure. Therefore, we took advantage of the MARTINI fixed secondary structure and tested both extremes that could happen in all-atom simulations. Moreover, these simulations allowed us to determine the role of the peptide flexibility (without altering the peptide-lipid interactions), which would be challenging to determine without simulations. Table 2 includes information about all simulated sequences and used secondary structures for the studied peptides.

Note that the secondary structure of antimicrobial peptides is highly dependent on the environment, and thus, the secondary structure from DPC micelles may not be representative. Furthermore, the secondary structure of peptides in solution, adsorbed on membrane, and in the pore can vary a lot. Consequently, it is challenging to determine/separate those structures in experiments. Nevertheless, we observed Magainin 2 (with a glycine kink) to form toroidal pores, which is consistent with the reviewer's comment and previous findings. In simulations with fully helical (without a kink), Magainin 2 forms a "synergistic" toroidal pore with two peptides interacting via C-termini (effectively forming a kink between two peptides).

In addition, we have included a comparison of the flexibility of both phenomenological and MARTINI models in Figure 10C in the Materials and methods section.

The experimental results in Figure 5 are valuable are limited to mutations at one residue (A) and seem to show no significant difference (this should be tested statistically) in (B)

We thank the reviewer for appreciation of our experiments. Since we already validated our results using two different computational models, we used only two peptides for experimental validation. We selected the peptides based on their availability and effect on membrane. The kink-forming residue has been the main subject of this study, thus in (A) we mutated residues at this position. We selected three different mutants P11L, P11A, and P11G to capture two variants of helix- (L, A) and kink- (P, G) forming residues. The results are in excellent agreement with our simulations and understanding. For LL-37, the induced leakage by the mutant peptide is within the error of the wild type. In this case, however, we cannot distinguish the effect of the peptide flexibility and the increased peptide-lipid interactions.

There should be some justification for the very large pores in Figure 6.

In our previous work (DOI: 10.1021/la402727a), we have shown that peptides can tilt inside a pore and even adopt a parallel orientation in the pore lumen resulting in fairly large pores.

Reviewer #2:[…] The weakness of the manuscript is in the description of the simulation work done. Thus, for example, to help the reader to understand the phenomenological model and the results from simulations with it, the authors should provide a brief description of this model, since, as I suspect the model used may be not as familiar to the reader as MARTINI force field. This description can be given in Supplementary Material. In some places it was hard for me to follow the paper because of its non-sequential order of referring to Figures and Tables (for example, see the Results were the reader is sent to figures 5D and S13 before discussion of previous figures). I also did not understand how many different initial configurations and arrangements were considered for each kind of peptide in MARTINI simulations. This is a very important issue, since simulations were done starting with preformed pores. The authors mention that the simulations were performed starting from different arrangements; see subsection “MARTINI simulations” and they refer to figure S1, that depicts a snapshot from simulation with phenomenological model. Also, it would be nice to know the length of each MARTINI simulation and the final outcome (pore morphology). Perhaps the authors can extend Table 1 to add useful information. Some of the Figures in Supplementary Material may be redundant. Thus, there is no reference in the text to Figures S3 and S4.

We thank the reviewer for the comments. We have rewritten the manuscript completely to improve the readability, and we addressed all the raised points. The phenomenological model description is expanded and we also added the Figure 10, which shows the model including the flexibility of the kink. Moreover, we described in detail how the initial pore structures were constructed including all tested initial configurations (Figure 11). For figures with representative pore structures from MARTINI simulations, we also added information about the robustness of the findings. Table A2 contains the number of independent simulations starting from different initial conditions leading to the same pore structure and the length of all simulations. For all studied peptides we include the information about the pore structure in Table 2. We sequential ordered Figures and Tables.

Summary: Interesting work, but the manuscript requires serious editing to streamline the arguments and make the paper more readable.Reviewer #3:The paper by Türkova et al. describes some low-resolution MD simulations of the effects of the kinks in AMPs on their pore-forming activity. The simulations give some interesting insights in particular as many interesting information can be obtained by very simple phenomenological models that merely represent hydrophobic / hydrophilic distributions. On the other hand, even the Martini model is probably not suitable to reproduce some experimental details (cf. below).Therefore, the authors need to be clearer about such details and discuss in depth a number of issues. Otherwise, the wrong conclusions can easily be drawn from the nice almost atomistic views shown in the paper.

We have rewritten the manuscript completely to improve the clarity and to address all the raised issues.

For example, it is of importance to underline that pore structures have been used as the starting structures for the simulations and that the analysis only compares which of the supposed models is more stable. A more convincing approach would have been to take a random distribution of peptides and check out if they assemble into stable or even transient pore structures, but this may not be possible even with the approaches choses that permit extended simulation times. As a matter of fact, all energies found for the various pore structures in the paper have positive energies and are unfavorable (when compared to the absence of a pore). As a consequence, none of the assumed models should represent the pore structures observed by experimental methods. This should be clearly mentioned and underlined. The paper would much increase in merit from such an open and unbiased discussion.

Antimicrobial peptides decrease the barrier for pore formation in a concentration-dependent manner. Above a critical concentration, the spontaneous pore formation is observable in experiments. In our previous work, we have shown that pore formation indeed becomes spontaneous at higher peptide-to-lipid ratios. However, the critical concentration is not known a priori. Moreover, it is usually more convenient to sample the whole free energy profile of pore formation in a sub-critical regime, which is why we have used it. Note that only lipids are biased towards the formation of a membrane pore, and the peptides can freely adopt the most favourable configuration. Importantly, the free energies, although positive, can be used to assess the effectiveness of the peptides.

The choice of the starting configurations is an important issue and we have used a few different configurations, which are displayed in Figure 10 and described in the Materials and methods section. We agree with the reviewer that observing spontaneous pore formation in MD simulations is generally not feasible within the accessible timescales and this is why we used the preformed pores.

Other models such as the carpet or the SMART model have not been discussed in the paper. There is good evidence from a number of experimental observations that neither the barrel stave nor the toroidal pore model represent well the activities of these peptides. The results of the paper seem to support such alternative views.

Carpet mechanism and detergent-like activity is a general behavior of amphiphilic peptides that occurs at sufficiently high concentrations, which sometimes even exceeding the lipid concentration. We have included references to other mechanisms of action. In the manuscript, we are interested in peptides that can form pores and are effective in membrane disruption at very low concentrations (P/L ration 1:100 or even lower).

As for the synergistic activities, similar considerations apply. An interesting observation is that in POPC or POPC/POPG membranes magainin and PGLa are both oriented parallel to the membrane surface in a stable manner. Only in membranes made of saturated lipids was membrane insertion of PGLa observed. It should be discussed to which extent the Martini model used here can represent these special properties of the lipid fatty acyl chains. Of course, it would be highly rewarding to model this different behavior of PGLa in the presence of magainin.

The behaviour of Magainin 2 and PGLa peptides is very interesting and complex, which can be demonstrated by a number of previously conducted studies providing a lot of data that cannot be easily reconciled. The investigation of such complex systems is beyond the scope of this study. Nevertheless, we are interested in Magainin 2 + PGLa system, and we have recently published the first paper of the series, which is focused on their synergistic behaviour DOI: 10.1016/j.bpj.2019.10.022.

In my opinion these are interesting data which should be presented in a different manner to further increase the merit of the paper.

We thank the reviewer for his interest, and we have rewritten the whole manuscript to improve the clarity and address all the issues.

[Editors’ note: what follows is the authors’ response to the second round of review.]

Reviewer #2:[…] The authors did a pretty good job of rewriting the manuscript. There are still small issues that make reading somewhat difficult in certain places: the reader needs to move forward from the main text to the section on Methods and Materials and back to understand clearly what is going on. A few sentences placed in the main text about methods can make the flow of reading much smoother. The authors should carefully read the manuscript and see how they can make the reading of the text smoother, so there is no need to go back and forth while reading the paper. Some very small remarks can be helpful; e.g., I got initially confused by notation used to denote figures such as Figure 1A and Figure A1. Just writing Figure A1 (see Appendix) can be very helpful.

We appreciate positive evaluation of our results, revision, and the suggestions to improve the text. We expanded the description of models and results to prevent the need to go back and forth in the text. We have also clarified a definition of bias applied on lipids in the Results section as recommended.

We changed notation of Figures in appendix, which are referenced as “Figure Apx1” now. This should prevent confusion with figures in the main part of the text.

Some other issues: 1) I did not understand what the free energy curve from Figure 1 denoted by dots stands for. 2) In subsection “Effect of Peptide Flexibility and Length on the Structure of Membrane Pores” the authors say that bias was applied to lipids to observe pore formation. What kind of bias?

1) Dotted lines in the Figure 1 and Appendix Figures Apx1 and Apx2 correspond to the free energy profile of pure membrane without any peptides. We added this information into the captions. 2) The bias was applied to the pore size (defined by pore area) and we describe this in the text.

Conclusion: The paper accomplished the goal of explaining the role of kinks as seen from the coarse-grained model and confirming leakage experiments. I recommend publication after some editing in response to remarks from above.Reviewer #4:I think most of the reviewer's comments have been adequately addressed. However, I do feel strongly that the authors should comment on their choice of membrane model and how it relates to in vivo compositions and any potential consequences of using such a highly charged membrane. Furthermore, the reviewers point about the justification of very large pores is not addressed to an appropriate level of detail. Justification for these pores should be clearly included in the manuscript.

Cellular membranes are composed of dozens of different lipids and the exact composition depends on a specific cell type, environment, or even diet. In order to make our results as general as possible, we studied simplified model membranes composed of the most common lipids. Zwitterionic POPC is typical and widely used lipid for single component membranes, which is why we used it in majority of our systems. However, we are aware that various lipids can have a specific role in different membranes. In particular, the negative charged lipids were suggested to be important for antimicrobial peptides. This is why we included membranes with negatively charged lipids in our study. In particular, we selected POPG lipid, which is common in bacteria, and POPS lipid, which is common in eukaryotic cells. To enhance the possible effect of charged lipids we employed 1:1 (mol/mol) lipid mixtures of POPC and charged lipid. The content of charged lipids may seem high, but even more highly charged membranes exist in bacteria. Moreover, antimicrobial peptides are expected to locally increase the concentration of negatively charged lipids because they are usually positively charged. Last but not least, lipid composition POPC:POPG 1:1 (mol/mol) was suggested as a suitable membrane model of bacteria based on the correlation of the leakage experiments and the minimum inhibitory concentration (MIC) measurements in live bacteria.

Large pores in our MARTINI simulations were used only in the initial stage of simulations, where peptides sampled various orientations and arrangements inside the pore. The large pores were used to allow easy reorientation of peptides into their preferred configuration.

We added the information in the manuscript.